# Hypomorphic *Brca2* and *Rad51c* double mutant mice display Fanconi anemia, cancer and polygenic replication stress

Karl-Heinz Tomaszowski[1], Sunetra Roy[1], Carolina Guerrero [1], Poojan Shukla[1], Caezaan Keshvani[1], Yue Chen[1], Martina Ott [2], Xiaogang Wu[3], Jianhua Zhang [3], Courtney D. DiNardo [4], Detlev Schindler [5] & Katharina Schlacher [1] ✉

The prototypic cancer-predisposition disease Fanconi Anemia (FA) is identified by biallelic mutations in any one of twenty-three *FANC* genes. Puzzlingly, inactivation of one *Fanc* gene alone in mice fails to faithfully model the pleiotropic human disease without additional external stress. Here we find that FA patients frequently display *FANC* co-mutations. Combining exemplary homozygous hypomorphic *Brca2/Fancd1* and *Rad51c/Fanco* mutations in mice phenocopies human FA with bone marrow failure, rapid death by cancer, cellular cancer-drug hypersensitivity and severe replication instability. These grave phenotypes contrast the unremarkable phenotypes seen in mice with single gene-function inactivation, revealing an unexpected synergism between *Fanc* mutations. Beyond FA, breast cancer-genome analysis confirms that polygenic *FANC* tumor-mutations correlate with lower survival, expanding our understanding of *FANC* genes beyond an epistatic FA-pathway. Collectively, the data establish a polygenic replication stress concept as a testable principle, whereby co-occurrence of a distinct second gene mutation amplifies and drives endogenous replication stress, genome instability and disease.

Genetic defects in *BRCA2* and *RAD51C* predispose to familial inherited breast and ovarian cancer. Biallelic inactivation of *BRCA2* (*FANCD1*) or *RAD51C* (*FANCO*) also causes Fanconi anemia (FA), an autosomal recessive disorder due to pathogenic mutations in one of twenty-three identified *FANC* genes. FA patients are small in stature and exhibit congenital abnormalities with skeletal deformations, malformations of inner organs, and impaired fertility[1]. FA entails bone marrow failure as well as hematological and solid cancers including breast cancer, typically at a young age[2,3]. In the context of breast cancer predisposition, *BRCA2* and *RAD51C* are well studied for their roles during homology-directed repair (HDR)[4]. The FA pathway goes beyond these functions, whereby all 23 genes are part of the DNA interstrand-crosslink repair pathway, a genetic tumor suppressor pathway. This pathway is specialized in the replication-coupled repair of particular DNA damage caused by free radicals including reactive oxygen species, aldehyde metabolites and chemotherapeutics that crosslink both DNA strands, such as mitomycin C and cisplatin drugs. By their very nature, FA patient cells are hyper-sensitive to pertinent chemotherapeutic toxicity, and show genomic and chromosomal instability that can drive resistance[3,5]. FANC pathway activation requires the sequential engagement of the FANC core complex, necessary for the mono-ubiquitination of the FANCD2/FANCI heterodimer, which then allows the recruitment of post-ID2 ubiquitination factors including nucleases, translesion synthesis components, and HDR proteins comprising BRCA1/2, RAD51, RAD51C[3]. Additionally, *FANC* genes also play an essential role in other processes during DNA duplication. Irrespective

[1]Department of Cancer Biology, The University of Texas MD Anderson Cancer Center, Houston, TX 77054, USA. [2]Department of Neurosurgery, The University of Texas MD Anderson Cancer Center, Houston, TX 77054, USA. [3]Department of Genomic Medicine, The University of Texas MD Anderson Cancer Center, Houston, TX 77054, USA. [4]Department of Leukemia, The University of Texas MD Anderson Cancer Center, Houston, TX 77054, USA. [5]Institut fuer Humangenetik, University of Wuerzburg, Wuerzburg, Germany. ✉e-mail: kschlacher@mdanderson.org

of the nature of the DNA damage that causes a roadblock for semi-conservative replication, FANC proteins promote DNA fork protection at stalled replication forks, which is a genome stability pathway that is genetically and mechanistically separable from DNA repair yet required during replication-coupled inter-strand crosslink repair[3,6–8]. Indeed, cells with specific patient mutations in either *BRCA2/FANCD1* or *RAD51/FANCR* are defective in replication fork protection but remain proficient for HDR[9,10], and restoration of fork protection without restoration of interstrand crosslink repair functions can reestablish hematopoietic stem cell functions in *Fancd2*[-/-] mice[11], suggesting physiological importance of this distinct molecular pathway in FA disease.

The more recent DNA sequencing efforts of tumors in patients without FA further underscore the importance of the FA pathway during tumor suppression and chemotherapy response, as somatic *FANC* gene mutations are commonly detected in diverse cancers, providing promise for tailored therapy targeting strategies[3,12]. The current paradigm is that knowing about *FANC* gene mutation status could be used as a genetic biomarker to personalize cancer treatment strategies as the mutations should render the repair deficient tumor cells sensitive to specific chemotherapeutics such as platinum drugs and PARP inhibitors. Thus, understanding *FANC* gene functions and the underlying biological causes that lead to the FA disease is poised to disentangle diverse aspects of cancer pathophysiology spanning from tumor suppression to chemotherapeutic agent efficacy.

Yet, developing fitting FA mouse models has been deemed challenging. While biallelic inactivation of any one of the twenty-three *FANC* genes serves as the defining clinical marker for the disease diagnostics[2,13,14], knock-out (KO) of any one *Fanc* gene does not cause FA in mice. Instead, *Fanc* KO mice exhibit select and mild manifestations of the disease and vary with the specific *Fanc* gene that is affected. Recently, improved FA disease modeling has been achieved by exposing the mice to additional stress in the form of externally applied DNA damaging agents or by increasing endogenous aldehydes by homozygous *Aldh2* KO in conjunction with ethanol treatment[15]. FA is also phenotypically heterogeneous in patients, with members of one family who share the same biallelic *FANC* mutation exhibiting greatly varying onset and severity of the disease[16]. It has been speculated that genetic background, modifier genes, chance events, variable teratogenic exposures during each pregnancy, and mosaicism could be potential contributors to observed pleiotropism[17].

We here show that combining homozygous hypomorphic *Brca2/Fancd1* and *Rad51c/Fanco* mutations in mice by polygenic crosses recapitulates a wide spectrum of the human FA disease, including congenital anomalies, hematopoietic malfunction, and cancer at a young age. In stark contrast to the polygenic mutant mice, mice with monogenic homozygous hypomorphic *Brca2* or *Rad51c* mutation alone express negligible phenotypes with no significant effect on the overall survival compared to wild-type mice. FA patient DNA analysis confirms that polygenic *FANC* mutations are a prevalent event in humans. The damaging effect of *FANC* co-mutations found in patients is supported by tumor DNA analysis of breast cancer patients without FA, who show significantly reduced survival with polygenic compared to monogenic *FANC* tumor mutations. Molecularly, the replication alteration caused by the *Rad51c* mutation amplifies the dysfunctional *Brca2* mutation resulting in hyperactive replication fork degradation and genome instability without external DNA damage. Thus, the data establishes the concept of polygenic replication stress where one gene mutation is insufficient for pathogenicity on its own, but requires distinct gene co-mutations that drive endogenous replication stress and genome instability. Collectively, the data highlights a non-epistatic FANC tumor-suppressor pathway and the importance of distinct supplemental stress in addition to a biallelic *FANC* mutation in the development of FA.

## Results

### Polygenic *FANC* mutations are frequent in human FA patients

FA is diagnosed by identifying disrupting mutations in one of twenty-three so far identified *FANC* genes. We performed a literature search and found multiple examples of FA patients with heterozygous or homozygous *FANC* gene mutations in addition to the inactivating *FANC* variants that the patient's disease is attributed to (Supplementary Table 1). To rigorously test for polygenic *FANC* mutations in FA patients, we performed whole-exome sequencing on DNA from 37 FA patients with known biallelic mutations in one given *FANC* gene. Strikingly, 27% of the patients tested harbor non-synonymous germline mutations in additional *FANC* genes (Fig. 1a and Supplementary Table 2), suggesting a high prevalence of polygenic heterozygous or homozygous *FANC* variants in FA patients.

### Polygenic *Brca2*[Δ27/Δ27] + *Rad51c*[dah/dah] mutations synergistically cause congenital defects in mice

Seemingly in contrast to the human diagnostic standards for FA, homozygous knock-outs or mutations in any one *Fanc* gene alone in mice do not cause FA, but instead only mild and selective disease phenotypes[18–20] that are exacerbated with exposure to stress[15,21,22]. The common occurrence of additional *FANC* mutations in FA patients led us to hypothesize that *Fanc* mutations in a second gene in addition to the inactivating biallelic *Fanc* mutation could have phenotypic relevance. We therefore sought to genetically test if combining *Fanc* gene mutations could provide an improved mouse model.

For this, we chose hypomorphic mutations in *Brca2* and *Rad51c* for the following reasons: First, we identified an FA patient with polygenic *BRCA2* + *RAD51C* mutations in the NIH dbGap database (Supplementary Table 3). Secondly, heterozygous hypomorphic germline mutations in *BRCA2* + *RAD51C* are also found in families with a history of breast cancer susceptibility[23]. Thirdly, FA patients' mutations frequently are small base pair-level variants[24–26], rather than gene knock-outs, and fourth, homozygous hypomorphic mutant *Brca2* mice are established (Fig. 1b, *Brca2*[Δ27/Δ27]). BRCA2 gene KO) is not viable in humans or mice, consistent with the gene's function in double-strand break repair that is essential to life[27,28]. While the BRCA2 C-terminus is necessary for fork protection, it is greatly dispensable for DNA break repair[10]. Some *BRCA2/FANCD1* FA patients contain C-terminal truncations (*BRCA2* p.Tyr3225fs*30)[25] almost identical to the homozygous hypomorphic mutation of the above mouse, making it a fitting model. Yet, mice with the homozygous C-terminal *Brca2* gene truncation[29] show mild phenotypes with somewhat reduced fertility and an increased risk of cancer with a late-age onset at >60 weeks[29], but not nearly FA to the extent seen in patients.

The *RAD51C/FANCO* mutations associated with FA patients (R258H[26] and G264S, Supplementary Table 3), are located within an α-helix, thus affecting the same structural unit (Supplementary Fig. 1a, b). Using CRISPR/Cas9 technology, we targeted this *Rad51c* α-helix in mice. The resulting homozygous *Rad51c/Fanco* mutant mice are viable and carry a six base-pair deletion mutation resulting in an internal truncation of the α-helix by deleting two leucines at positions 270/271, corresponding to human L261/L262 (*Rad51c*[dah/dah]; "dah" denotes "deletions within α-helix"; Fig. 1c, d and Supplementary Fig. 1a–e). The deletion is sandwiched between the FA/cancer patient-associated RAD51C amino acids R258H and G264S, corresponding to murine R267 and G273 (Supplementary Fig. 1a). A two amino-acid deletion within an α-helix causes a rotation of the helix and affects the location, and thus likely the function of the disease suppressing amino acids (Supplementary Fig. 1b). In contrast to *Rad51c* KO mice[27], the mutant *Rad51c*[dah/dah] mice express the internally-deleted protein (Fig. 1e), supporting their viability.

As the best approximation to patient *FANC* mutations, we crossed these hypomorphic *Rad51c* and *Brca2* mice. Polygenic mutant *Brca2*[Δ27/Δ27] + *Rad51c*[dah/dah] mice are viable, but the

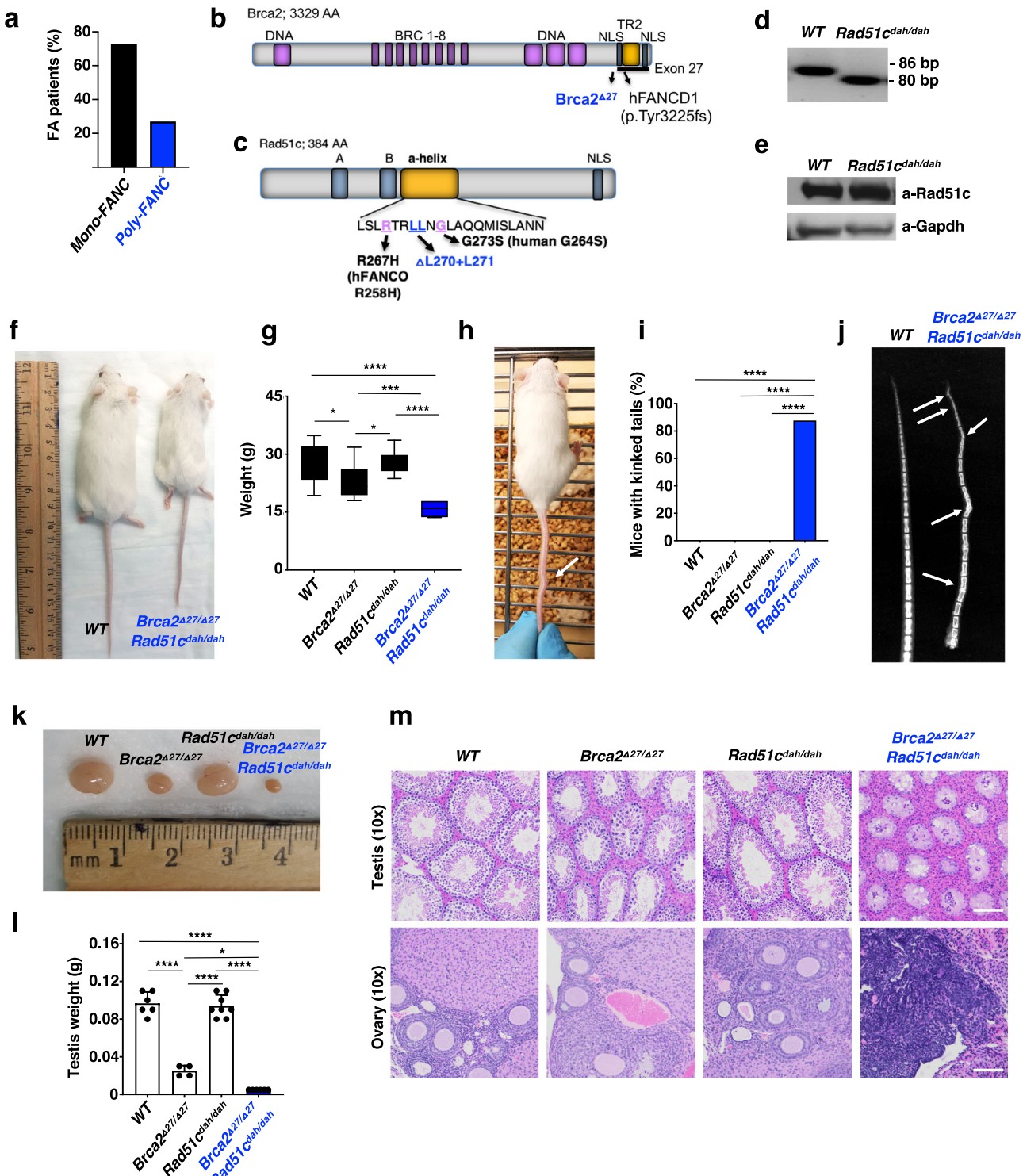

frequency from polygenic heterozygous intercrosses is two-fold to two-and-a-half-fold lower compared to the expected Mendelian birth rate ($p = 0.01$ and $0.001$, respectively; Supplementary Table 4).

FA patients frequently are born with congenital abnormalities causing small stature and skeletal malformations[2,13,14]. In contrast to the monogenic homozygous mutant mice, $Brca2^{\Delta 27/\Delta 27}$ + $Rad51c^{dah/dah}$ mice are very small and have a significantly reduced weight, which is readily discernable from an early age (Fig. 1f, g). Additionally, $Brca2^{\Delta 27/\Delta 27}$ + $Rad51c^{dah/dah}$ mice, but not single-mutant $Rad51c^{dah/dah}$ or $Brca2^{\Delta 27/\Delta 27}$ mice, exhibit visible skeletal deformations with kinked tails (Fig. 1h, i).

We further examined the tails by X-ray imaging to determine the cause of the tail kinks and found that many of the tail vertebrae in $Brca2^{\Delta 27/\Delta 27}$ + $Rad51c^{dah/dah}$ mice are fused and exhibit an abnormal, thickened vertebrae shape (Fig. 1j), similar to vertebral malformations seen with the VACTERL association, which is often connected with FA in humans[24,30].

Congenital developmental defects can cause loss of fertility, often observed in FA patients[31]. We examined the primary sex organs in the mice and found that while the testis of $Brca2^{\Delta 27/\Delta 27}$ mice are smaller compared to $Rad51c^{dah/dah}$ mice or wild-type mice, the testis size in mice with polygenic $Brca2^{\Delta 27/\Delta 27}$ + $Rad51c^{dah/dah}$ mutations is further

**Fig. 1 | Polygenic $Brca2^{\Delta27/\Delta27}$ + $Rad51c^{dah/dah}$ mutant mice show congenital deformations. a** Exome sequencing of 37 confirmed FA patient serum filtered for additional passenger mutations defined as non-synonymous FANC mutations in additional FANC genes other than the primary, diagnostic biallelic FANC mutation. **b** Schematic of murine Brca2. Depicted are the BRCA2/FANCD1 mutation identified in the FANCD1 patient (c.9672dupA), which results in a truncation of exon 27 (p.Tyr3225fs*30). The $Brca2^{\Delta27/\Delta27}$ mouse bears an exon 27 deletion likewise truncating Brca2 similar to the human FANCD1 truncation. AA, amino acids; BRC, Rad51 binding motifs; TR2, Rad51 stabilization domain; DNA, DNA binding folds/motifs; NLS, nuclear localization sequence. **c**, Schematic of murine Rad51c. Depicted in orange is the α-helix that contains the RAD51C/FANCO mutation identified in the FANCO patient (human R258H, murine R268), and the adjacent site for the two-amino acid deletion in the $Rad51c^{dah/dah}$ mouse. "dah" denotes "deletions within a-helix"; AA, amino acids; A,B, Walker A and B motifs; NLS, nuclear localization sequence. **d** PCR genotyping of ear biopsies showing the six base pair (bp) deletion. PCR genotyping were performed >100 times with similar results. **e** Western blot of wild-type (WT) and $Rad51c^{dah/dah}$ mouse testis cell extracts showing similar protein expression. Experiment was performed once. **f** Representative image of wild-type and $Brca2^{\Delta27/\Delta27}$ + $Rad51c^{dah/dah}$ mice at 12 weeks exhibiting noticeable growth defect. **g** Box-whisker plot of weight of mice with indicated genotypes at 12 weeks. Box centers indicate median, boundaries represent 25th and 75th percentiles, and error

bars represent maximum and minimum values. Data represent biologically independent samples from wild-type ($n = 18$), $Brca2^{\Delta27/\Delta27}$ ($n = 20$), $Rad51c^{dah/dah}$ ($n = 7$), and $Brca2^{\Delta27/\Delta27}$ + $Rad51c^{dah/dah}$ ($n = 8$) mice. $Brca2^{\Delta27/\Delta27}$ + $Rad51c^{dah/dah}$ against WT and $Rad51c^{dah/dah}$: $p < 0.0001$; $Brca2^{\Delta27/\Delta27}$ + $Rad51c^{dah/dah}$ against $Brca2^{\Delta27/\Delta27}$: $p = 0.0005$; WT against $Brca2^{\Delta27/\Delta2}$: $p = 0.018$; $Brca2^{\Delta27/\Delta27}$ against $Rad51c^{dah/dah}$: $p = 0.0181$. **h** Representative image of $Brca2^{\Delta27/\Delta27}$ + $Rad51c^{dah/dah}$ mouse with kinked tail (arrow). **i** Box plot of frequency of mice with kinked tails with indicated genotypes ($n = 10$ mice). **j** Representative X-ray image of tails of WT and $Brca2^{\Delta27/\Delta27}$ + $Rad51c^{dah/dah}$ mice indicating skeletal abnormalities (arrows). **k** Representative image of testis of mice with indicated genotypes at 12 weeks. **l** Bar graph of weight of testis with indicated genotypes at 12 weeks. Data represent biologically independent samples from wild-type ($n = 6$), $Brca2^{\Delta27/\Delta27}$ ($n = 4$), $Rad51c^{dah/dah}$ ($n = 8$) and $Brca2^{\Delta27/\Delta27}$ + $Rad51c^{dah/dah}$ ($n = 3$) mice. $Brca2^{\Delta27/\Delta27}$ + $Rad51c^{dah/dah}$ against WT and $Rad51c^{dah/dah}$: $p < 0.0001$; $Brca2^{\Delta27/\Delta27}$ + $Rad51c^{dah/dah}$ against $Brca2^{\Delta27/\Delta2}$: $p = 0.0194$; WT against $Brca2^{\Delta27/\Delta27}$: $p < 0.0001$; $Brca2^{\Delta27/\Delta27}$ against $Rad51c^{dah/dah}$: $p < 0.0001$. **m** Representative images of hematoxylin and eosin (H&E)-stained tissue cross-sections of testis and ovary, respectively, from animals with indicated genotypes at 12 weeks. Scale bars indicate 100 µm. Similar results were obtained by analyzing three independent tissues of each genotype. Error bars represent the standard error of the mean. $p$-values are derived using the one-way ANOVA test. *$p < 0.05$, ***$p < 0.001$, ****$p < 0.0001$.

---

significantly reduced (Fig. 1k, l). Histological examination of testis cross-sections using hematoxylin and eosin staining shows that the seminiferous tubuli in $Rad51c^{dah/dah}$ mice testis appear normal, while $Brca2^{\Delta27/\Delta27}$ mice show a greater heterogeneity accompanied by some mice expressing a mild decrease in spermatocytes residing in the lumen of the tubuli (Fig. 1m and Supplementary Fig. 2a), consistent with the mild fertility defect previously reported in this mouse model[29]. In stark contrast, the polygenic $Brca2^{\Delta27/\Delta27}$ + $Rad51c^{dah/dah}$ mice show a complete loss of spermatogonia germ cells in addition to all spermatocytes (Fig. 1m, upper panels, and Supplementary Fig. 2a). The data uncovers a Sertoli-cell only phenotype in these mice, which is a common FA patient phenotype[31] and causes complete infertility. Similarly, cross-sections of ovaries show the complete loss of ovarian follicles in the $Brca2^{\Delta27/\Delta27}$ + $Rad51c^{dah/dah}$ mice, but not in $Rad51c^{dah/dah}$, $Brca2^{\Delta27/\Delta27}$, or wild-type mice (Fig. 1m, lower panels, and Supplementary Fig. 2b, c). Collectively, the polygenic but not monogenic mutant mice show severe congenital development defects resembling those seen in FA patients.

## Polygenic $Brca2^{\Delta27/\Delta27}$ + $Rad51c^{dah/dah}$ mutations cause strong hematopoietic defects

FA frequently leads to bone marrow failure, characterized by a hypocellular bone marrow with progressive peripheral pancytopenia resulting in low blood cell counts. We first counted the number of bone marrow cells, which are lowest in $Brca2^{\Delta27/\Delta27}$ + $Rad51c^{dah/dah}$ mice (Fig. 2a). To further test if any blood cells are preferentially affected, we performed a blood cell analysis by complete blood counts on young mice (age 8–12 weeks, Fig. 2b–e, and Supplementary Fig. 3a–h). While the number of platelets appears similar in all genotypes at this age (Supplementary Fig. 3a), there is a small but consistent decrease in the white blood cells in the $Brca2^{\Delta27/\Delta27}$ + $Rad51c^{dah/dah}$ mice compared to the single mutant mice (Fig. 2c). More specifically, differentiated cells from the myeloid, lymphoid and erythroid lineages are reduced in the polygenic mutant mice, including monocytes, granulocytes, leukocytes, and red blood cells (Fig. 2c, d and Supplementary Fig. 3b–f). Early red cell aplasia is accompanied by macrocytosis, which is one of the earliest hematological disease indicators in FA[32]. Consistent with the manifestations in FA patients, $Brca2^{\Delta27/\Delta27}$ + $Rad51c^{dah/dah}$ mice but not $Rad51c^{dah/dah}$ nor $Brca2^{\Delta27/\Delta27}$ mice show significantly enlarged red blood cells, suggesting macrocytosis as a sign of reduced or delayed hematopoiesis (Fig. 2e and Supplementary Fig. 3h). Collectively, the data obtained from differentiated cells of myeloid, erythropoietic

and lymphatic lineages in young polygenic $Brca2^{\Delta27/\Delta27}$ + $Rad51c^{dah/dah}$ mutant mice is consistent with early bone marrow failure.

To further trace the cells of origin for the hematological defect, we performed hematopoietic lineage flow cytometric analysis[33]. Consistent with the reduced counts in differentiated cells, the bone marrow of $Brca2^{\Delta27/\Delta27}$ + $Rad51c^{dah/dah}$ mice contains less of both, myeloid and common lymphoid progenitor cells, albeit the numbers do not allow for statistical significance in the young mice (Fig. 2f, g and Supplementary Fig. 4a, b). Intriguingly, the frequency of the hematopoietic stem cells (HSC) on the other hand is slightly increased in polygenic compared to single mutant mice at the time-point measured (Fig. 2h). Increased HSCs are a sign of stress-induced HSC proliferation, which can ultimately lead to their early depletion[34]. We therefore sought to further test bone marrow cell functions. Colony-forming unit assays revealed a functional defect of $Brca2^{\Delta27/\Delta27}$ + $Rad51c^{dah/dah}$ bone marrow cells, albeit $Brca2^{\Delta27/\Delta27}$ proliferation-potent cells too show a reduction in colony formation capacity (Fig. 2i). To stringently test HSC function, we performed a bone marrow repopulation assay by injecting an equal amount of bone marrow cells from either male $Brca2^{\Delta27/\Delta27}$ + $Rad51c^{dah/dah}$ and female wild-type mice, or male $Brca2^{\Delta27/\Delta27}$ and female wild-type mice into irradiated, bone marrow depleted mice (Fig. 2j, k). Strikingly, quantifying the frequency of Y-chromosome DNA in blood cells 16 weeks after transplantation we found that while the bone marrow of $Brca2^{\Delta27/\Delta27}$ mice significantly contributes to the make-up of the blood cells in the experimental animals, $Brca2^{\Delta27/\Delta27}$ + $Rad51c^{dah/dah}$ bone marrow cells completely fail to do so (Fig. 2k). Thus, the data demonstrates a dramatic loss of HSC function in vivo in $Brca2^{\Delta27/\Delta27}$ + $Rad51c^{dah/dah}$ bone marrow.

## Polygenic $Brca2^{\Delta27/\Delta27}$ + $Rad51c^{dah/dah}$ cells show hyper-replication instability and aggravated sensitivity to chemotherapeutics

The FA pathway suppresses several hallmarks of cancer, most prominently genome instability upon DNA damage. FANC proteins are critical for the specialized repair of DNA crosslink damage and the stabilization of stalled replication forks. Replication stress also drives hematopoietic dysfunction[35], and hematopoietic cell function in $Fancd2^{-/-}$ mice can be rescued by restoring DNA fork protection, a distinct form of DNA fork stabilization[11]. We therefore next focused on the replication fork function of the $Brca2$ and $Rad51c$ genes and performed single-molecule DNA fiber spreading to assess DNA replication fork protection in mouse adult fibroblasts (MAF) isolated from monogenic and polygenic mutant mice. Even without external replication stress, the DNA fibers of the polygenic $Brca2^{\Delta27/\Delta27}$ + $Rad51c^{dah/dah}$

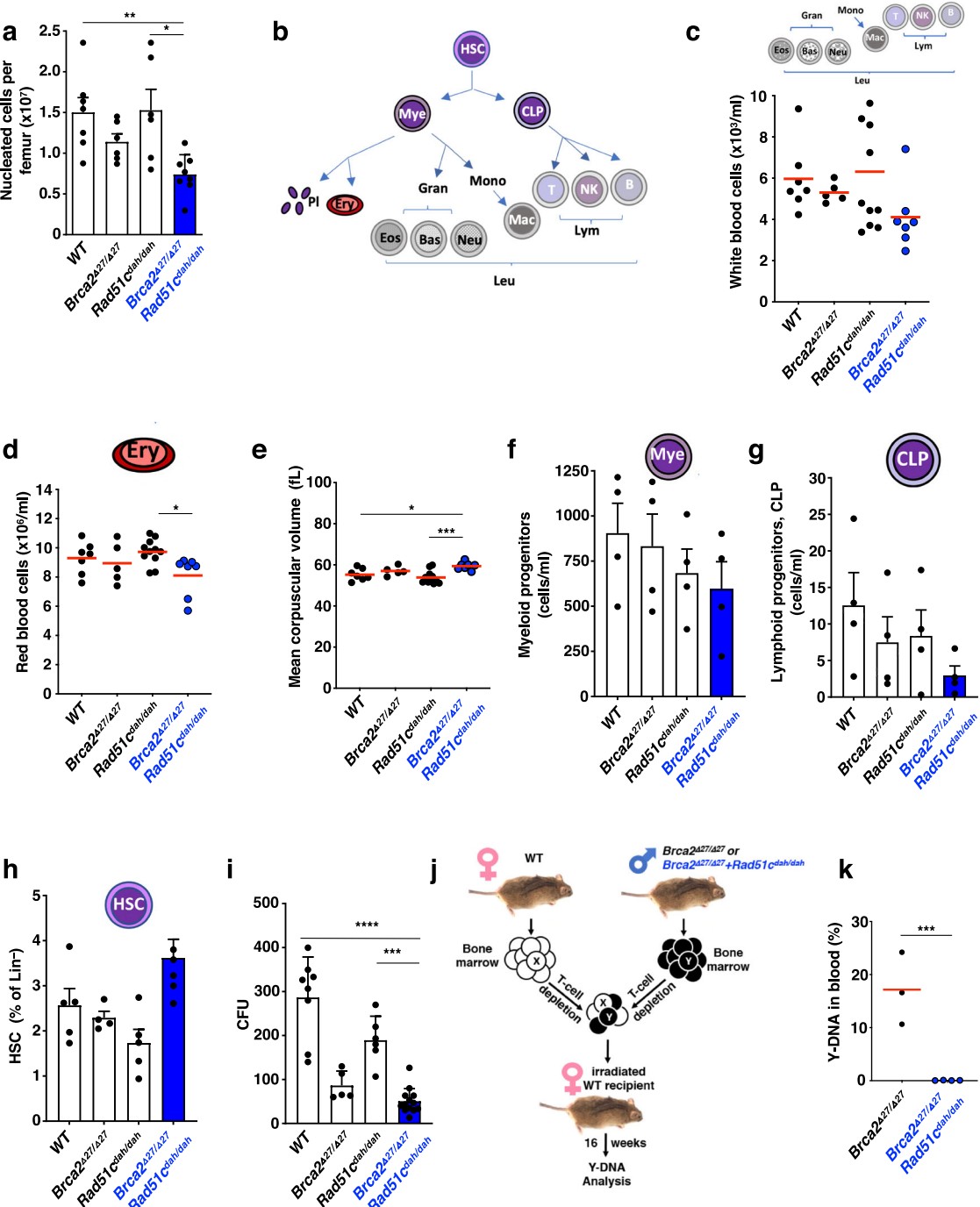

MAFs show the shortest nascent tracts, indicating higher intrinsic replication stress (Supplementary Fig. 5a).

To gain a better understanding of the particular replication reactions, we assessed both DNA replication restart and DNA fork protection capacities in these cells. Replication restart is the ability to resume replication after stalling, while replication fork protection is the suppression of degradation of the nascent DNA that was synthesized before replication stalling[10,36]. Previous reports showed that while RAD51C is involved in both replication restart and fork protection, BRCA2 plays a role during fork protection only and is dispensable for replication restart[10,37]. Consistently, we find that *Brca2^{Δ27/Δ27}* MAFs show a considerable fork protection defect upon replication stalling with high concentrations of hydroxyurea (HU, Fig. 3a), but are

proficient for replication restart (Fig. 3b). *Rad51c^{dah/dah}* MAFs on the other hand show a considerably milder fork protection defect with high concentrations of HU (Fig. 3a), but in contrast to the *Brca2^{Δ27/Δ27}* MAFs exhibit a significant defect in replication restart (Fig. 3b). *Brca2^{Δ27/Δ27} + Rad51c^{dah/dah}* MAFs show the strongest fork protection defect, which can be partially rescued by overexpressing wild-type Rad51c in these cells, suggesting that both *Rad51c* and *Brca2* contribute to the observed replication instability phenotype (Supplementary Fig. 5b, c). With mild external replication stress upon low concentrations of HU, there is little degradation of nascent strands in either the *Brca2^{Δ27/Δ27}* MAFs or *Rad51c^{dah/dah}* MAFs (Fig. 3c). In stark contrast, the *Brca2^{Δ27/Δ27} + Rad51c^{dah/dah}* MAFs show significantly shorter replication tracts even under these mild conditions (Fig. 3c). Taken together, the

**Fig. 2 | Hematological analysis of young polygenic $Brca2^{\Delta27/\Delta27}$ + $Rad51c^{dah/dah}$ mutant mice reveals pre-state of bone-marrow failure. a** Bar graph of nucleated cells per femurs of mice between 8–12 weeks. Data represent biologically independent samples from wild-type ($n = 7$), $Brca2^{\Delta27/\Delta27}$($n = 6$), $Rad51c^{dah/dah}$ ($n = 6$) and $Brca2^{\Delta27/\Delta27}$ + $Rad51c^{dah/dah}$ ($n = 8$) mice. $Brca2^{\Delta27/\Delta27}$ + $Rad51c^{dah/dah}$ against WT: $p = 0.0099$; $Brca2^{\Delta27/\Delta27}$ + $Rad51c^{dah/dah}$ against $Rad51c^{dah/dah}$: $p = 0.0106$. **b** Schematic of hematological cell lineages. HSC, hematopoietic stem cell. Mye, myeloid progenitor. CLP, common lymphoid progenitor. Pl, platelets. Ery, Erythrocytes, red blood cells. Leu, Leukocytes, white blood cells. Gran, Granulocytes. Eos, Eosinophils. Bas, Basophiles. Neutr, Neutrophils. Mono, Monocytes. Macro, Macrophages. Lym, Lymphocytes. T, T-cells. B, B-cells. NK, natural killer cells. **c–e** Complete blood count analysis of 8–12 weeks old $Brca2^{\Delta27/\Delta27}$ + $Rad51c^{dah/dah}$ and control mice. Data represent biologically independent samples from wild-type ($n = 7$), $Brca2^{\Delta27/\Delta27}$($n = 5$), $Rad51c^{dah/dah}$ ($n = 11$) and $Brca2^{\Delta27/\Delta27}$ + $Rad51c^{dah/dah}$ ($n = 7$) mice. **c** Peripheral white cell concentration. **d** Peripheral red cell concentration. $Brca2^{\Delta27/\Delta27}$ + $Rad51c^{dah/dah}$ against $Rad51c^{dah/dah}$: $p = 0.0271$. **e** Mean corpuscular volume as a measure for macrocytosis. $Brca2^{\Delta27/\Delta27}$ + $Rad51c^{dah/dah}$ against WT: $p = 0.0321$; $Brca2^{\Delta27/\Delta27}$ + $Rad51c^{dah/dah}$ against $Rad51c^{dah/dah}$: $p = 0.0005$. **f–i** Hematological lineage analysis by flow cytometry was performed for bone marrow cells collected from 8–12-weeks-old mice with indicated genotypes. **f** Bar graph of common myeloid progenitor population. Data represent 4 biologically independent samples for each genotype. **g** Bar graph of common lymphoid progenitor population. Data represent 4 biologically independent samples for each genotype. **h** Bar graph of hematopoietic stem cell population. Data represent biologically independent samples from wild-type ($n = 5$), $Brca2^{\Delta27/\Delta27}$($n = 4$), $Rad51c^{dah/dah}$ ($n = 5$) and $Brca2^{\Delta27/\Delta27}$ + $Rad51c^{dah/dah}$ ($n = 6$) mice. **i** Bar graph of colony formation capacity (CFU) of bone marrow progenitor cells per 30.000 plated cells. Data represent independent samples for wild-type ($n = 8$), $Brca2^{\Delta27/\Delta27}$($n = 5$), $Rad51c^{dah/dah}$ ($n = 6$) and $Brca2^{\Delta27/\Delta27}$ + $Rad51c^{dah/dah}$ ($n = 13$). $Brca2^{\Delta27/\Delta27}$ + $Rad51c^{dah/dah}$ against WT: $p < 0.0001$; $Brca2^{\Delta27/\Delta27}$ + $Rad51c^{dah/dah}$ against $Rad51c^{dah/dah}$: $p = 0.0002$. **j** Schematic of 16-week repopulation assay workflow. Briefly, bone marrow cells from female wild-type (WT) and male mutant mice were mixed and allowed to repopulate in female mice that were depleted for their bone-marrow. After 16 weeks, the repopulation capacity of mutant bone marrow is measured by PCR of male cells in the blood, which were contributed by the mutant mouse. **k** Quantitation of repopulation assay. Error bars represent the standard error of the mean. Red bar in scatter graphs represent the mean. Data represent biologically independent animals with $Brca2^{\Delta27/\Delta27}$($n = 3$) and $Brca2^{\Delta27/\Delta27}$ + $Rad51c^{dah/dah}$ ($n = 4$) genotypes. $Brca2^{\Delta27/\Delta27}$ against $Rad51c^{dah/dah}$: $p = 0.0006$. $p$-values (A-I) are derived using the one-way ANOVA test. $* p < 0.05$, $*** p < 0.001$, $**** p < 0.0001$. The $p$-values (K) are derived using the two-tailed Student $t$ test. $*** p < 0.001$.

data shows that cells with polygenic *Fanc* mutations show intrinsically augmented replication stress that amplifies the DNA replication instability with otherwise insignificant replication stalling.

High genome instability is also linked to hypersensitivity to DNA crosslinking agents, which is a key feature of FA patients' cells[3,5], and on the flip side of the coin renders the disease a valuable natural model system to understand *FANC* mutation-related drug sensitivity. Using a colony formation assay, we see that bone marrow cells from $Brca2^{\Delta27/\Delta27}$ + $Rad51c^{dah/dah}$ mice show severe cellular sensitivity to mitomycin C (MMC) on survival, consistent with an FA cell profile (Fig. 3d). In comparison, cells from $Rad51c^{dah/dah}$ or $Brca2^{\Delta27/\Delta27}$ mice exhibit only very mild to moderate sensitivity (Fig. 3d). Metaphase spreads of primary ear fibroblasts exposed to a high concentration of MMC (20 ng/ml) show increased chromosomal aberrations in both $Rad51c^{dah/dah}$ and $Brca2^{\Delta27/\Delta27}$ fibroblasts compared to wild-type fibroblasts. In contrast, $Brca2^{\Delta27/\Delta27}$ + $Rad51c^{dah/dah}$ fibroblasts had significantly more aberrations including radial formation than any of the other genotypes, which are partially repressed by transient expression of wild-type Rad51c (Fig. 3e–g and Supplementary Fig. 6a, b). In agreement with the DNA fiber results, we observed that radial structures were less pronounced in $Brca2^{\Delta27/\Delta27}$ fibroblasts at a lower concentration of MMC (2 ng/ml), compared to what is seen in $Brca2^{\Delta27/\Delta27}$ + $Rad51c^{dah/dah}$ fibroblasts, suggesting that the polygenic mutant mice are substantially more sensitized even at low amounts of DNA damage (Fig. 3e, f). Without any external genotoxic stress, neither $Rad51c^{dah/dah}$ nor $Brca2^{\Delta27/\Delta27}$ fibroblasts showed any more measurable aberrations compared to wild-type fibroblasts (Fig. 3h, i and Supplementary Fig. 6c). Yet, spontaneous chromosomal aberrations are significantly increased in $Brca2^{\Delta27/\Delta27}$ + $Rad51c^{dah/dah}$ fibroblasts, suggesting strong intrinsic genomic vulnerability of the polygenic FANC mutant cells.

### Polygenic *Brca2/Rad51c* mutations cause T-cell leukemia in mice

We next determined the overall survival of polygenic mutant FA mice. Dramatically, all $Brca2^{\Delta27/\Delta27}$ + $Rad51c^{dah/dah}$ mice die between 3-4 months of age (87-120 days, Fig. 4a). This is in stark contrast to wild-type and either of the single-gene mutant mice, where neither $Rad51c^{dah/dah}$ nor the $Brca2^{\Delta27/\Delta27}$ mice show any significant difference in overall survival in the first year of life.

Patients with FA related to defects of BRCA2/FANCD1 exhibit a variety of cancers, including T-cell leukemia, acute myeloid leukemia, medulloblastoma, Wilms and other embryonal-type tumors[3,24,38]. We found that the polygenic mutant mice develop strikingly enlarged thymus and spleen (Fig. 4b and Supplementary Fig. 7a), suggestive of

hematologic malignancy. Moreover, leukemic mice have an unusually elevated number of white blood cells (Supplementary Fig. 7b), and blood smears show the occurrence of blast cells with a uniform appearance (Fig. 4c, white arrows). Additional hematopoietic lineage analysis of $Brca2^{\Delta27/\Delta27}$ + $Rad51c^{dah/dah}$ mice by flow cytometry revealed that thymus cells are greatly enlarged in the mutant mouse (Supplementary Fig. 7c), and exhibit strongly abnormal T-cell differentiation (Fig. 4d). Furthermore, lymphatic tissues including the bone marrow, spleen and blood all show a large number of CD4 + /CD8 + progenitor T-cells (Fig. 4e, f), but not B-lymphocytes or myeloid cells (Supplementary Fig. 7d, e), indicating the presence of lymphoblastic T-cell leukemia (T-ALL).

Neoplasms in FA patients are characterized by high genomic instability. To further analyze leukemia arising in $Brca2^{\Delta27/\Delta27}$ + $Rad51c^{dah/dah}$ mice we assessed chromosome copy numbers and T-cell clonality using next-generation whole-exome sequencing. Copy number variations were greatly enriched in the leukemia samples compared to non-malignant ear samples (Fig. 4g). Moreover, by analyzing the frequency of T-cell receptors α and β, we found that $Brca2^{\Delta27/\Delta27}$ + $Rad51c^{dah/dah}$ mice show oligo- or monoclonal T-cell expansion (Fig. 4h, i). The overall frequency of T-cell clones was strongly reduced in T-ALL mice compared to control samples. Together, the analysis suggests that the polygenic $Brca2^{\Delta27/\Delta27}$ + $Rad51c^{dah/dah}$ mice die of T-cell leukemia that, in contrast to classic T-ALL, shows a high degree of intrinsic genomic instability, consistent with the characteristics of FA patient malignancies.

### Polygenic *FANC* mutations in breast tumors are associated with significantly lower survival of breast cancer patients

Germline mutations in certain FA genes predispose patients to cancer including breast cancer[3]. The polygenic mouse model revealed a stark mutual reinforcement between germline mutations in *Brca2* and *Rad51c* with increased genome instability and substantially worse organismal outcome. We therefore asked whether this phenomenon extended beyond FA and tested if somatic polygenic *RAD51C* + *BRCA2* mutations in tumors of cancer patients without FA could similarly show functional synergism. We analyzed the overall survival of patients with tumors harboring *BRCA2*, *RAD51C* or combinations of *BRCA2* + *RAD51C* mutations using the breast cancer genome datasets available at cBioportal.org[39,40]. cBioportal defines mutations as reported by the original study and includes non-synonymous mutations, amplifications and deletions (Supplementary Table 5). As any of these mutations can cause functional gene

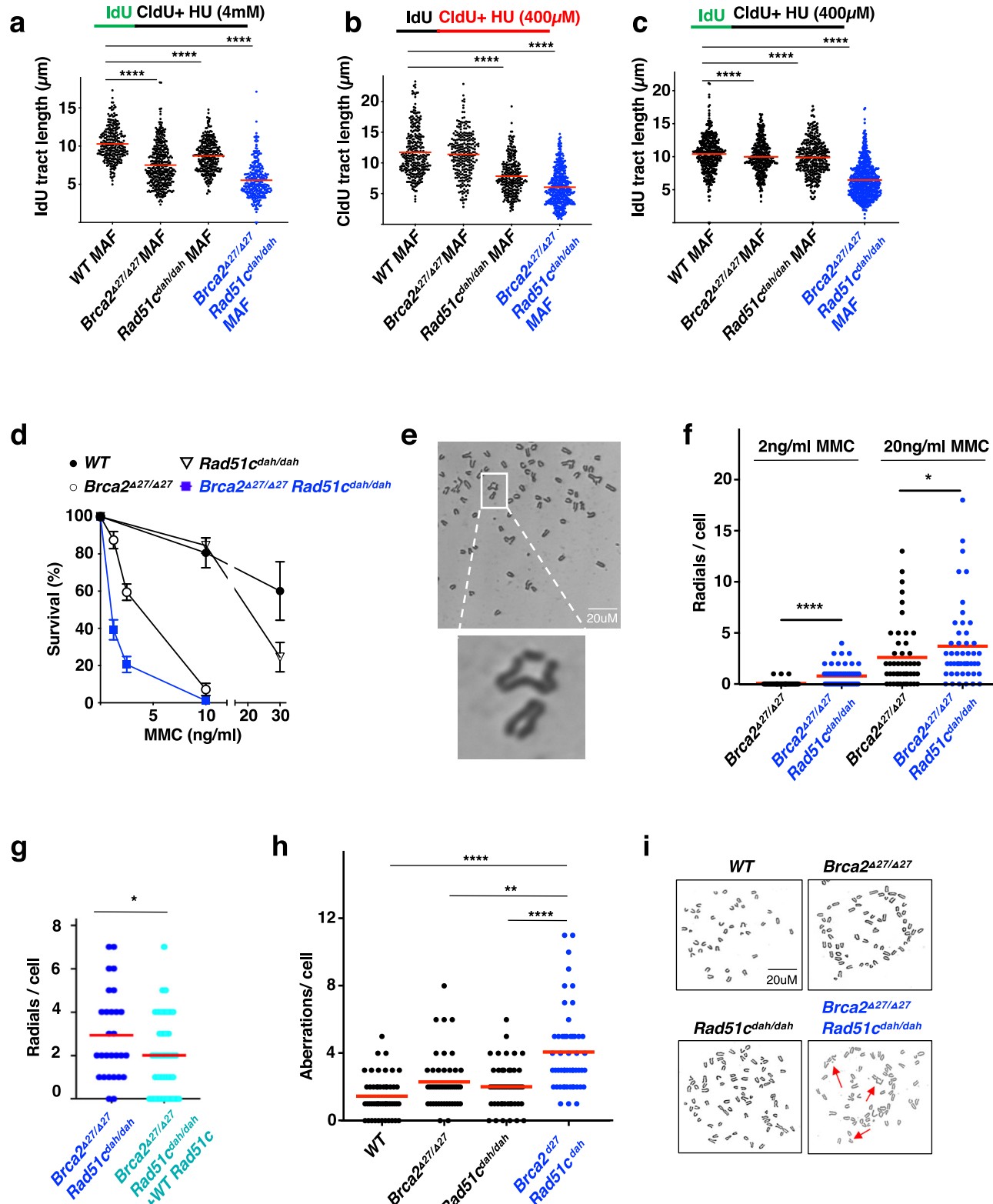

defects, we remained agnostic to the type of mutation and solely assessed the genetic association of polygenic compared to monogenic *BRCA2* and *RAD51C* mutation with the overall survival of the patient as a phenotypic outcome and endpoint. Strikingly, polygenic mutations in *BRCA2 + RAD51C* in breast tumors led to significantly more severe disease and reduced the overall survival by >90 months compared to patients with single *BRCA2* or *RAD51C* mutations only (Figs. 5a, 3.2-fold decrease in median overall survival in breast cancer patients with polygenic compared to monogenic *BRCA2* or *RAD51C* tumor mutations, 3.5-fold decrease in median survival in patients with polygenic tumor mutations compared to patients with no *BRCA2* or *RAD51C* mutation). In contrast, monogenic *BRCA2* or *RAD51C* tumor mutations confer only a minor but insignificant decrease in overall survival compared to patients with no *BRCA2* or *RAD51C* mutation (Fig. 5a), revealing genetic vulnerability associated with the polygenic *BRCA2 + RAD51C* mutations.

**Fig. 3 | Polygenic *Brca2*$^{\Delta 27/\Delta 27}$ + *Rad51c*$^{dah/dah}$ mutations confer hyper-DNA replication and genome instability. a** Scatter plot of DNA fiber analysis of nascent IdU tract lengths incorporated before replication stalling with high concentrations of hydroxyurea (HU, 4 mM), measuring DNA fork protection. Data represent biologically independent samples from wild-type (*n* = 3), *Brca2*$^{\Delta 27/\Delta 27}$ (*n* = 2), *Rad51c*$^{dah/dah}$ (*n* = 3) and *Brca2*$^{\Delta 27/\Delta 27}$ + *Rad51c*$^{dah/dah}$ (*n* = 2) measurement. In total, analysis was performed for wild-type (*n* = 293), *Brca2*$^{\Delta 27/\Delta 27}$ (*n* = 399), *Rad51c*$^{dah/dah}$ (*n* = 319) and *Brca2*$^{\Delta 27/\Delta 27}$ + *Rad51c*$^{dah/dah}$ (*n* = 306) fibers. *Brca2*$^{\Delta 27/\Delta 27}$, *Rad51c*$^{dah/dah}$ and *Brca2*$^{\Delta 27/\Delta 27}$ + *Rad51c*$^{dah/dah}$ against WT: *p* < 0.0001. **b** Scatter plot of DNA fiber analysis of nascent CldU tract lengths incorporated with replication stalling with low concentrations of hydroxyurea (HU, 400 μM), measuring replication restart. Data represent biologically independent samples from wild-type (*n* = 4), *Brca2*$^{\Delta 27/\Delta 27}$ (*n* = 3), *Rad51c*$^{dah/dah}$ (*n* = 3), and *Brca2*$^{\Delta 27/\Delta 27}$ + *Rad51c*$^{dah/dah}$ (*n* = 3) measurement. In total, analysis was performed for wild-type (*n* = 303), *Brca2*$^{\Delta 27/\Delta 27}$ (*n* = 263), *Rad51c*$^{dah/dah}$ (*n* = 282), and *Brca2*$^{\Delta 27/\Delta 27}$ + *Rad51c*$^{dah/dah}$ (*n* = 495) fibers. *Rad51c*$^{dah/dah}$ and *Brca2*$^{\Delta 27/\Delta 27}$ + *Rad51c*$^{dah/dah}$ against WT: *p* < 0.0001. **c** Scatter plot of DNA fiber analysis of nascent IdU tract lengths incorporated before replication stalling with low concentrations of hydroxyurea (HU, 400 μM), measuring DNA fork protection at low stress conditions. Data represent biologically independent samples from wild-type (*n* = 4), *Brca2*$^{\Delta 27/\Delta 27}$ (*n* = 3), *Rad51c*$^{dah/dah}$ (*n* = 3) and *Brca2*$^{\Delta 27/\Delta 27}$ + *Rad51c*$^{dah/dah}$ (*n* = 3) measurement. In total, analysis was performed for wild-type (*n* = 410), *Brca2*$^{\Delta 27/\Delta 27}$ (*n* = 329), *Rad51c*$^{dah/dah}$ (*n* = 337) and *Brca2*$^{\Delta 27/\Delta 27}$ + *Rad51c*$^{dah/dah}$ (*n* = 658) fibers. *Brca2*$^{\Delta 27/\Delta 27}$ against WT: *p* = 0.0103, *Rad51c*$^{dah/dah}$ against WT: *p* = 0.005 and *Brca2*$^{\Delta 27/\Delta 27}$ + *Rad51c*$^{dah/dah}$ against WT: *p* < 0.0001. **d** Cellular sensitivity of bone marrow cells to varying concentrations of Mitomycin C (MMC) as indicated. The number of colonies of progenitor cells were determined after 14 days. Data represent biologically independent bone marrow samples from wild-type (*n* = 3), *Brca2*$^{\Delta 27/\Delta 27}$ (*n* = 3), *Rad51c*$^{dah/dah}$ (*n* = 3), and *Brca2*$^{\Delta 27/\Delta 27}$ + *Rad51c*$^{dah/dah}$ (*n* = 4) mice. **e** Representative images of metaphase chromosome spreads in *Brca2*$^{\Delta 27/\Delta 27}$ + *Rad51c*$^{dah/dah}$ MAFs. Similar structure designed as radial were observed in >100 independent samples. **f** Scatter dot blot of radial chromosome structures in metaphase spreads of mouse adult fibroblasts (MAF) after treatment with 2 or 20 ng/ml MMC as indicated. Data represent 3 independent experiments. In total, chromosome analysis was performed for 2 ng/ml MMC treated *Brca2*$^{\Delta 27/\Delta 27}$ (*n* = 48) and *Brca2*$^{\Delta 27/\Delta 27}$ + *Rad51c*$^{dah/dah}$ (*n* = 86) cells as well as for 20 ng/ml MMC treated *Brca2*$^{\Delta 27/\Delta 27}$ (*n* = 56) and *Brca2*$^{\Delta 27/\Delta 27}$ + *Rad51c*$^{dah/dah}$ (*n* = 48) cells. 2 ng/ml MMC-treated *Brca2*$^{\Delta 27/\Delta 27}$ + *Rad51c*$^{dah/dah}$ against *Brca2*$^{\Delta 27/\Delta 27}$: *p* < 0.0001; 20 ng/ml MMC-treated *Brca2*$^{\Delta 27/\Delta 27}$ + *Rad51c*$^{dah/dah}$ against *Brca2*$^{\Delta 27/\Delta}$: *p* = 0.0575; **g** Scatter dot blot of radial chromosome structures (20 ng/ml MMC) with or without transient expression of wild-type Rad51c. Data represent 2 independent experiments. In total, chromosome analysis was performed for GFP-transfected *Brca2*$^{\Delta 27/\Delta 27}$ + *Rad51c*$^{dah/dah}$ (*n* = 28) cells and for Rad51c-transfected *Brca2*$^{\Delta 27/\Delta 27}$ + *Rad51c*$^{dah/dah}$ (*n* = 41) cells. *Brca2*$^{\Delta 27/\Delta 27}$ + *Rad51c*$^{dah/dah}$ against *Brca2*$^{\Delta 27/\Delta 27}$ + *Rad51c*$^{dah/dah}$ + WT *Rad51c* *p* = 0.0608. **h** Scatter dot blot of spontaneous chromosome aberrations in metaphase spreads in MAFs. Data represent 3 independent experiments. In total, chromosome analysis was performed for untreated wild-type (*n* = 50), *Brca2*$^{\Delta 27/\Delta 27}$ (*n* = 52), *Rad51c*$^{dah/dah}$ (*n* = 46), and *Brca2*$^{\Delta 27/\Delta 27}$ + *Rad51c*$^{dah/dah}$ (*n* = 53) cells. *Brca2*$^{\Delta 27/\Delta 27}$ + *Rad51c*$^{dah/dah}$ against WT and *Rad51c*$^{dah/dah}$: *p* < 0.0001; *Brca2*$^{\Delta 27/\Delta 27}$ + *Rad51c*$^{dah/dah}$ against *Brca2*$^{\Delta 27/\Delta 27}$: *p* = 0.002. **i** Representative images of metaphase chromosome spreads without exogenous damage. Similar pictures of metaphases with total number of aberrations for each genotype were observed in >10 independent samples. Error bars represent the standard error of the mean. Bars represent the mean of compiled data from biological repeats. *p*-values for DNA fiber analysis and the genomic instability analysis with or without transient expression of wild-type Rad51c (**g**) are derived using the two-tailed Mann–Whitney test, and the *p*-values for the genomic instability analysis (**f**, **h**) are derived using the one-way Anova test. *$p$ < 0.1, **$p$ < 0.01, ***$p$ < 0.001, ****$p$ < 0.0001.

We further tested if the effect of polygenic mutations extends to *FANC* genes other than *BRCA2* and *RAD51C*. For this we focused on the altogether sixteen *FANC* gene mutation combinations that we identified to be present in FA patients through literature searches (Supplementary Table 1) and in a diagnostic setting (Supplementary Table 2). We first compared the overall survival of breast cancer patients with tumors without *FANC* mutations (Fig. 5b, "NO FANC") to those harboring single mutations in any one of the *FANC* genes (Fig. 5b, "MONOGENIC FANC"). The data shows that tumors with monogenic *FANC* mutations are not *per se* associated with more adverse patient survival outcome compared to tumors that have no *FANC* mutation. In contrast, the overall survival of breast cancer patients with tumors harboring polygenic *FANC* mutations in these sixteen FA patient-associated gene pairs show a significantly reduced overall survival by >55 months compared to patients with monogenic *FANC*-gene mutations only (Fig. 5b, "FA patient POLYGENIC FANC"). The data is consistent with the second mutation promoting a more adverse outcome when in combination with the other *FANC* gene defect in the FA patient-associated gene pairs. Of note, while the pathological make-up of breast cancer subtypes observed with polygenic *FANC* mutations is similar compared to those with single mutations (Supplementary Fig. 8a), these tumors are genetically more unstable as seen by a significantly increased count of somatic mutations throughout the genome in the polygenic *FANC* tumors (Supplementary Fig. 8b).

While we so far identified sixteen polygenic *FANC* mutation combinations in FA patients, there are 253 possible combinations between all twenty-three *FANC* genes. Importantly, when querying all 253 combinations, effectively there is no difference in the overall survival between patients with tumors containing polygenic, monogenic or no *FANC* mutations (Fig. 5c and Supplementary Fig. 8c). Taken together, the data shows not all polygenic *FANC* mutation combinations per se are associated with lower survival in breast cancer patients. Instead it supports the notion that the polygenic *FANC* mutations identified in FA patients may not have been found by chance but have functional consequences in a non-epistatic manner.

## Discussion

Clinically, FA is caused by inactivation of one of 23 so far identified *FANC* genes[2,13,14]. Of note, Guido Fanconi, the Swiss pediatrician after whom the disease is named, speculated that the disease was too complex to be caused by one gene[41]. We here show that polygenic, double-homozygous *Brca2*$^{\Delta 27/\Delta 27}$ + *Rad51c*$^{dah/dah}$ gene mutations in mice closely recapitulate the FA disease manifestations found in patients, providing a comprehensive preclinical cancer mouse model of FA.

Similar to other *Fanc* mutant mouse models, monogenic homozygous *Brca2*$^{\Delta 27/\Delta 27}$ or *Rad51c*$^{dah/dah}$ mutations on their own do not elicit profound FA phenotypes in mice. Interestingly, externally applied agents that cause DNA damage and DNA replication stress by aldehydes[15] or DNA crosslinks[21,22] in monogenic *Fanc* mutant mice much better model the human FA disease manifestations compared to the single *Fanc* gene alterations alone. We find that cells from the polygenic mutant mice show significantly worse DNA replication instability. While the *Brca2*$^{\Delta 27/\Delta 27}$ mice also are fork protection defective to some degree, they are proficient in restarting stalled forks. We suggest that the additional restart defect conferred by the *Rad51c* mutation in the polygenic mice is one means of causing sufficient internal replication stress in this mouse model to promote unchecked replication fork degradation and replication hyper-instability. Thus, the genetic combination of *Fanc* inactivation could cause endogenous stress, in analogy to the principle of oncogene-induced replication stress[42]. In support of this idea, cells from polygenic mutant mice exhibit significantly higher spontaneous chromosomal aberrations, while the single mutant mouse cells only show increased genomic instability compared to wild-type cells with exogenously provoked DNA damage. FA genes act in many biological processes, most notably during various DNA repair processes, so the phenotypes in the mice could in principle stem from defects other than those involving their direct function in replication reactions. Nevertheless, DNA damage ultimately too results in DNA replication stress and instability. Our proposed concept of "polygenic replication stress" by compounding genetic alterations resulting in strong DNA replication instability furthermore is affirmed by the observation that the same *Fanc* gene

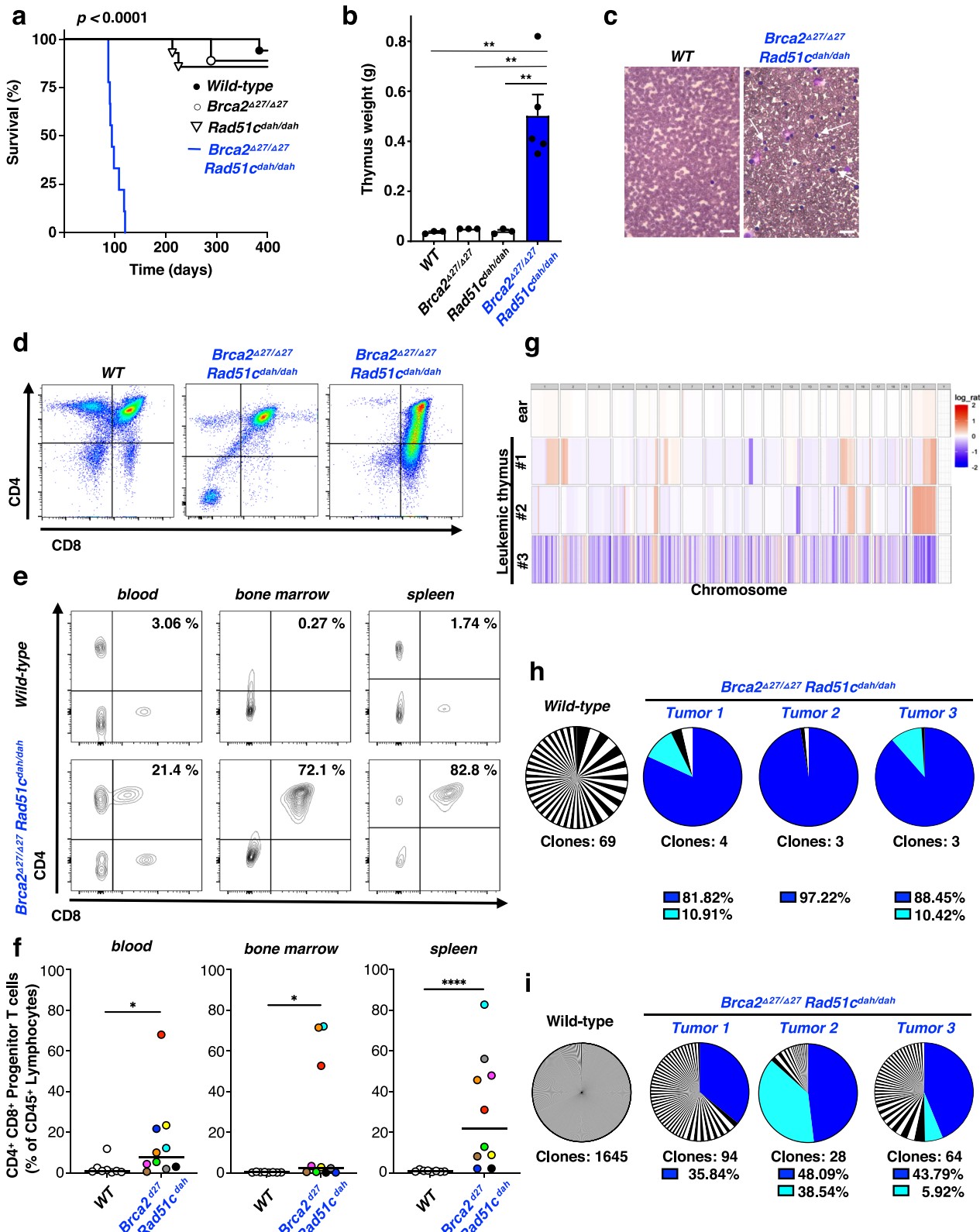

knockouts cause varying phenotypic severity depending on the genetic background of the purebred animals[19,43,44], with the gravest phenotypes seen in the C57BL strain backgrounds. Indeed, we find that MAFs from C57BL/ 6NCrl mice but not other mouse strain backgrounds show an intrinsic fork protection defect (Supplementary Fig. 9a), which could explain the more severe phenotype of a particular *Fanc* gene inactivation in this but not other mouse strains.

It previously was reported that combining FANCD2 with BRCA1 or BRCA2 depletions can cause increased fork protection defects in cells[45,46]. However, the dramatic physiological effects of polygenic *Fanc* gene inactivation as seen in our mouse model reveal an unexpected synergism of genetic interactions within the FA tumor suppressor pathway that shapes disease manifestations and the response to chemotherapeutics. Indeed, polygenic *FANC* tumor mutations in

**Fig. 4 | Polygenic *Brca2^{Δ27/Δ27}* + *Rad51c^{dah/dah}* mutations are synergistic during cancer development. a**, Kaplan-Meier curves for overall survival of wild-type (WT) (*n* = 17), *Brca2^{Δ27/Δ27}* (*n* = 9), *Rad51c^{dah/dah}* (*n* = 14), and *Brca2^{Δ27/Δ27}* + *Rad51c^{dah/dah}* (*n* = 9) mice. *Brca2^{Δ27/Δ27}* + *Rad51c^{dah/dah}* against all other genotypes: *p* < 0.0001; WT against *Brca2^{Δ27/Δ2}*: *p* = 0.9890; WT against *Rad51c^{dah/dah}*: *p* = 0.4957; *Brca2^{Δ27/Δ27}* against *Rad51c^{dah/dah}*: *p* = 0.8058. *p*-values for comparing survival of two groups are derived using the Mantel-Cox test, **b**, Bar-graph of thymus mass quantitation in malade *Brca2^{Δ27/Δ27}* + *Rad51c^{dah/dah}* and age-matched control mice. Data are presented as mean values ± SEM and represent biologically independent samples for wild-type (*n* = 3), *Brca2^{Δ27/Δ27}* (*n* = 3), *Rad51c^{dah/dah}* (*n* = 3) and *Brca2^{Δ27/Δ27}* + *Rad51c^{dah/dah}* (*n* = 5) mice. *Brca2^{Δ27/Δ27}* + *Rad51c^{dah/dah}* against WT: *p* = 0.0018; *Brca2^{Δ27/Δ27}* + *Rad51c^{dah/dah}* against *Brca2^{Δ27/Δ2}*: *p* = 0.0022; *Brca2^{Δ27/Δ27}* + *Rad51c^{dah/dah}* against *Rad51c^{dah/dah}*: *p* = 0.0019. **c** Representative images of blood films (×40 magnification) obtained from malade *Brca2^{Δ27/Δ27}* + *Rad51c^{dah/dah}* mice and age-matched WT mice shows high number of abnormal leukocytes (arrow). Scale bar, 30 μm. Similar results with high number of abnormal leukocytes were observed in three independent samples. **d** Flow cytometry analysis of thymus cells from malade *Brca2^{Δ27/Δ27}* + *Rad51c^{dah/dah}*

mice (middle and right) show abnormal development of T-cells (low mature CD4 + or mature CD8 + T-cells), indicating the development of acute lymphoblastic leukemia. **e, f** Flow cytometry analysis of blood, bone marrow and spleen cells from malade *Brca2^{Δ27/Δ27}* + *Rad51c^{dah/dah}* mice show high frequency of progenitor T-cells (double CD4 + CD8 + T-cells) in various lymphatic tissues, indicating acute lymphoblastic leukemia. Data represent biologically independent samples for wild-type (*n* = 8) and *Brca2^{Δ27/Δ27}* + *Rad51c^{dah/dah}* (*n* = 10) mice. *Brca2^{Δ27/Δ27}* + *Rad51c^{dah/dah}* against WT for blood: *p* = 0.0117; for bone marrow: *p* = 0.016; and for spleen *p* < 0.0001. **g** Somatic copy number aberration analysis of three leukemia thymus samples and one ear tissue in *Brca2^{Δ27/Δ27}* + *Rad51c^{dah/dah}* mice. **h** TCRβ clone frequencies identified by whole-exome sequencing of thymus cells from wild-type and three leukemic *Brca2^{Δ27/Δ27}* + *Rad51c^{dah/dah}* mice. **i** TCRα clone frequencies identified by whole-exome sequencing of thymus cells from wild-type and three leukemic *Brca2^{Δ27/Δ27}* + *Rad51c^{dah/dah}* mice. *p*-values for flow cytometry analysis are derived using the two-tailed Mann–Whitney test, and the *p*-values for thymus mass analysis are derived using the one-way Anova test. * *p* < 0.1, ** *p* < 0.01, *** *p* < 0.001, **** *p* < 0.0001.

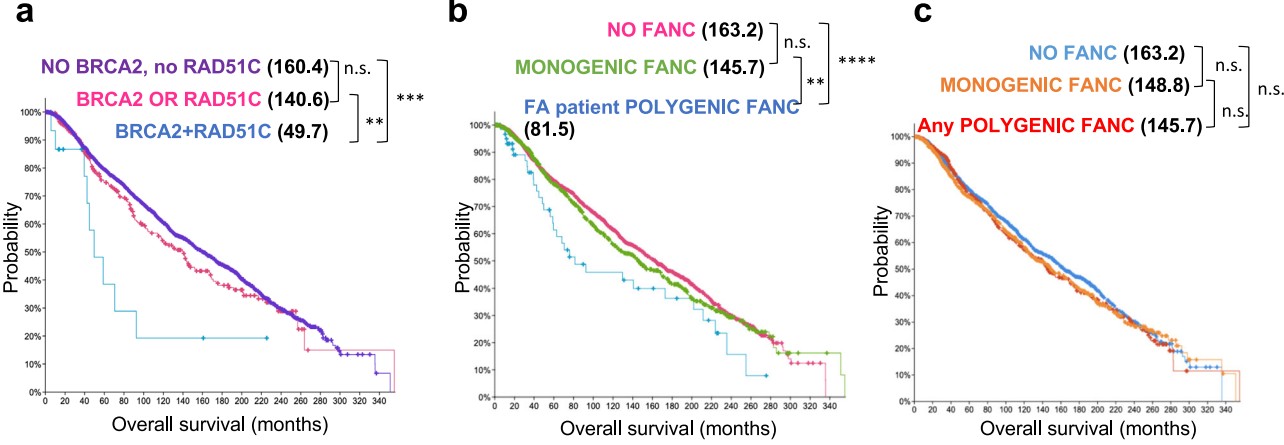

**Fig. 5 | cBioportal analysis reveals polygenic *FANC* tumor mutation combinations found in FA patients are associated with decreased survival in breast cancer patients. a** Kaplan–Meier curves for overall survival of breast cancer patients with tumor mutations in both, BRCA2 and RAD51C (polygenic BRCA2 + RAD51C, *n* = 16, *p* = 0.00611), with mutations in either, BRCA2 and RAD51C genes (monogenic BRCA2 + RAD51C, *n* = 318, *p* = 0.224), or with no BRCA2/RAD51C tumor mutations (*n* = 2729). **b** Kaplan-Meier curves for overall survival of breast cancer patients with tumor mutations in combinations of FANC genes identified to be

mutated in the same FA patients (16 pairs, FA patient polygenic FANC, *n* = 63, *p* = 0.009663), with mutations in one FANC gene (monogenic FANC, *n* = 923, *p* = 0.212), or with no FANC tumor mutations (*n* = 2077). **c** Kaplan–Meier curves for overall survival of breast cancer patients with tumor mutations in any combinations of FANC genes identified in FA patients (Any polygenic FANC, *n* = 623), with mutations in one FANC gene (monogenic FANC, *n* = 796), or with no FANC tumor mutations (*n* = 1644, *p* = 0.216). *p*-values for all survival curves are derived from cBioportal using the Log-Rank test.

breast cancer patients are also associated with significantly decreased overall survival, suggesting that polygenic *FANC* mutations in humans have physiological relevance beyond FA also in cancer. Importantly, the genetic interactions are not random and do not occur with any *FANC* gene combination. Thus, the data implies that FA can inform of previously unappreciated synthetic lethal interactions exploitable for new therapeutic strategies.

FA occurs with a frequency of 1:100,000–250,000. Inconsistently, ~4.3% of individuals in the Western population carry heterozygous disease-causing *FANC* variants, making the disease far less frequent than what would be expected from Mendelian transmission[47,48]. In recent years it has become evident that there is a proportion of patients that present only in adulthood, which may have skewed these estimates to some extent. On the other hand, the requirement for stress factors in addition to biallelic *FANC* mutations too could account in part for this discrepancy. This additional stress can include physical DNA damage caused by endogenous or exogenous aldehyde damage, such as previously seen in *Fancd2^{-/-}* + *Aldh2^{-/-}* mice[15]. While there are reports of FA patients with heterozygous *ALDH2* co-mutations, phenotypic manifestations in mice appear to require homozygous *Aldh2* mutations[49]. Consistent with this observation, we find predominantly

heterozygous co-mutations in humans, while the polygenic homozygous mice show the most severe phenotypes. This seeming discrepancy could perhaps reflect the sterile living environment of laboratory animals. Indeed, we find that heterozygous co-mutations accelerate phenotypes when exposed to extrinsic damage such as gamma irradiation (Supplementary Fig. 9b).

Regardless, the data presented here suggests that endogenous replication stress by additional *Fanc* gene mutations is sufficient to cause the disease phenotypes in mice. The concept of polygenic *Fanc* interactions beyond an epistatic FA repair pathway is consistent with previously inexplicable epistasis studies in *Fanc* KO mice, whereby *Fancc^{-/-}* + *Fancg^{-/-}* but not *Fancc^{-/-}* + *Fanca^{-/-}* mice showed a worsening of the otherwise mild phenotypes[43,50]. As *FANC* genes are increasingly found to participate in non-canonical repair and replication processes[51,52], it will be of great interest to comprehensively unravel the specific causes and functional defects of unexpected FANC interactions in future studies. Moreover, the additional synergistic mutations may not be restricted to so far identified *FANC* genes but feasibly can include other genetic mechanisms that induce stress. In support of this idea, *Fancc^{-/-}* + *Sod1^{-/-}* mice with the latter gene mutation promoting oxidative stress show a much improved FA disease modeling over the

monogenic *Fanc* model[53], as do *Fancd2^{-/-}+Aldh2^{-/-}* mice[15] demonstrating the broader importance of polygenic mutations in FA development beyond Brca2 and Rad51c. The data here show that polygenic *FANC* mutations are a frequent event in FA patients, together lending support to the concept of polygenic replication stress as an alternative mechanism to induce FA disease manifestations. It further suggests that additional gene analysis in FA patients could aid in predicting the severity of the disease as an added parameter, which is of particular importance in families with clinically heterogeneously affected siblings[54].

The concept of requiring polygenic gene inactivation for disease manifestation may usefully inform other rare genetic disease models beyond FA. In an as yet experimentally untested mathematical model, the co-recessive inheritance theory by Lambert and Lambert predicted that diseases caused by mutations in DNA damage response genes may require more than one gene inactivation[55], including ataxia telangiectasia (AT, associated with mutations in *ATM*), xeroderma pigmentosum and Cockayne syndrome (XP and CS, involving nucleotide excision repair genes), and cancer. A requirement of polygenic stress by synergistic gene mutations may at least in part resolve why single-gene inactivation alone only poorly models AT and CS in mice resulting in mild symptoms, respectively, while additional gene inactivation results in better disease modeling[56,57]. This phenomenon was so far unexplained. Thus, polygenic stress defining disease presentation as postulated, based on the observations of our mouse model, may point to a general concept.

Notably, polygenic risk evaluation, whereby the risk scores of multiple genes are added up to a compounded risk factor, is emerging as a promising tool in predicting many rare genetic diseases, including neurodevelopmental disorders, autism, Alzheimer's disease, and breast cancer[58–61]. Yet challenges remain in defining proper predictive parameters. The polygenic replication stress concept presented here predicts that the promising but so far incomplete success of polygenic risk prediction can be improved by considering the non-random and synergistic nature of the *FANC* mutation combinations presented here. The data here further has additional implications for tumor passenger mutation, whereby a seemingly unremarkable mutation in one gene is asymptomatic and tolerated, similar to the mutation in the monogenic *Rad51c^{dah/dah}* mice. However, within the context of an additional mild somatic mutation, represented in our mouse model by the *Brca2^{Δ27/Δ27}* mutation, it causes severe and aggressive cancer. Collectively, the presented mouse model offers a comprehensive preclinical model to faithfully investigate diverse FA disease manifestations and establishes the concept of polygenic replication stress synergism driving disease severity and therapy response as a testable principle for disease prognosis and new synthetic lethal cancer-therapeutic strategies.

## Methods

### Institutional review board statement
All procedures and methods were conducted in accordance with federal and state regulations as well as MD Anderson Cancer Center institutional guidelines and policies, as all procedures performed on animals were described in an Animal Care and Use Form (ACUF) and approved by the institutional animal care and use committee at the MD Anderson Cancer Center (IACUC).

### Mouse models
*Rad51^{dah/dah}* mice were generated by pronuclear injection using the CRISPR/Cas9 genome editing system. Cas9 and sgRNA 5'-CTTCGTACTCGATTACTAAATGG-3' were injected into pronuclear stage mouse embryos from FVB/J mice to generate founder animals. *Brca2^{Δ27/Δ27}* mice, carrying a C-terminal truncation, were obtained from NCI Mouse Repository (SWR.129P2(Cg)-*Brca2^{tm1Kamc}*/Nci, Strain code: 01XG9, SWR.129P2). Mice carrying both the *Brca2* C-terminal deletion and the *Rad51c* 6 base pair deletion were generated by

crossing *Brca2* and *Rad51c* heterozygous mutant mice. The obtained offspring were subsequently intercrossed to obtain double-mutant mice and control genotypes in FVB/J SWR.129 hybrid background. Mice were maintained in pathogen-free conditions. Animal experiments were performed per approved animal protocol by the Institutional Animal Care and Use Committee of the University of Texas M. D. Anderson Cancer Center.

For genotyping, the ears of mice were clipped and genomic DNA was extracted using DirectPCR Lysis Reagent (Viagen Biotech). PCR amplification was performed using GoTaq Green Master Mix (Promega) or Phusion High-Fidelity DNA polymerase (NEB) according to the manufacturer's introduction. PCR products (Rad51c-Forward, 5'-CGTCATGACCTTGAAGATC-3', Rad51c-Reverse, 5'-GATTATTTGCAAGGCTGATC-3'; Brca2-Forward, 5'-GAGAGCCCCATGCAGCCTCCACTTGCTGTG-3', and Brca2-Reverse, 5'-CTGCCTCCAGAGACCTGAGCCGTC-3') were directly analyzed by agarose gel based on product size.

### Generation of mouse adult fibroblasts
Primary ear fibroblasts were derived from age-matched males and females. Briefly, a portion of the ear was cut off, rinsed two times with PBS containing kanamycin (100 µg/mL) and digested with collagenase D/dispase II protease (4 mg/mL, respectively) for 45 min at 37 °C. After dilution with five times Dulbecco's modified eagle's high-glucose media (DMEM) containing 10% fetal bovine serum (FBS) and 5% Antibiotic-Antimycotic, cells were incubated overnight at 37 °C. The following day, cells were passed through a 0.7 µM cell strainer, washed with PBS and plated for cultivation in standard media consisting of DMEM supplemented with 10% FBS and 100 units/ml Pen-Strep. Mouse adult fibroblasts (MAFs) were grown at 37 °C and 5% $CO_2$, routinely passaged two times per week and passages 2–10 were used for experiments.

### Transfection of mouse adult fibroblasts
For transfection, $2.5 \times 10^5$ mouse adult fibroblasts were plated for 48 h in standard DMEM media into 6-well plates. Afterward, cells were transfected with 1 µg of Rad51c-GFP-expressing vector (NM_053269, OriGene) using Lipofectamine 3000 (Invitrogen) according to the manufacturer's instructions.

### Cytogenetic analysis
For chromosomal aberration analysis, $5 \times 10^4$ mouse adult fibroblasts were plated for 24 h in standard DMEM media. On the next day, exponentially growing cells were treated with the indicated concentration of mitomycin C for 15 h followed by incubation with colcemid (0.1 mg/ml, Gibco) for 5 h. Afterwards, cells were swollen with 0.04% KCl solution (12 min, 37 °C), fixed in methanol/acetic acid (3:1), dropped onto microscope slides, stained with 5% Giemsa solution and directly imaged with a Nikon Eclipse Ti-U inverted microscope. Images were analyzed using ImageJ software.

### Histological analysis
Testis and ovaries from adult mice were resected and fixed in 10% buffered formalin solution (Thermo Fisher Scientific) overnight and stored in 70% ethanol. Formalin-fixed tissues were paraffin-embedded and tissue sections were counterstain in eosin-phloxine B solution for 30 s to 1 min. Images were obtained using a Nikon Eclipse Ti-U inverted microscope.

### Bone marrow harvest
Bone marrow cells were isolated from femurs of 8–12-weeks-old mutant mice and appropriate controls. Skin and muscle surrounding the bone were removed and bone marrow cells were isolated by flushing out cells with PBS using a 27 gauge needle. Pelleted cells were treated with red cell lysis buffer (Lonza) for two minutes to eliminate red blood cells.

## Flow cytometry

Flow cytometry was performed on freshly isolated bone marrow cells. The following antibodies were used to stain for hematopoietic stem cells (HSCs) and progenitor populations: biotin-conjugated lineage cocktail with antibodies anti-TER-119, anti-CD11b, anti-Ly-6G/Ly-6C (Gr-1), anti-CD3e and anti-CD45R/B220 (BioLegend), anti-streptavidin (APC-Cy7, BD Pharmingen) was used as the secondary antibody, anti-c-kit (APC, clone 2B8, BD Pharmingen), anti-Sca-1 (PerCP-Cy5.5, clone D7, eBioscience), anti-CD34 (FITC, clone RAM34, eBioscience), anti-CD135 (Flt3) (PE, clone A2F10, eBioscience), and anti-CD127 (PE-Cy7, clone SB/199, eBioscience). For analyzing progenitor and mature blood cells, single cell suspension from the thymus and spleen were obtained by mashing the tissue through a 0.45 μM cell strainer and removing red blood cell using red cell lysis buffer. Bone marrow cells were harvested as described above. Afterward, cells were stained using a cocktail with the following antibodies: anti-CD3 (APC, clone 145-2C11, BD Pharmingen), anti-CD4 (PE, clone RM4-5, BD Pharmingen), anti-CD8 (FITC, clone 53-6.7, BD Pharmingen), anti-Gr-1 (unconjugated, BioLegend), anti-CD45 (PerCP-Cy5.5, clone 30-F11, BioLegend), anti-B220 (PE-Cy7, clone RA3-6B2, BioLegend), and anti-CD11b (BV605, clone M1/70, BioLegend). The samples were incubated with primary antibodies for 90 min at 4 °C in the dark. Following the washing steps, cells were incubated with secondary antibody anti-streptavidin (APC-Cy7, BD Pharmingen) for 30 min at 4 °C in the dark. For quantification of cell numbers, AccuCheck counting beads (Thermo Fisher Scientific) were included in samples and calculation was performed following the manufacturer's instructions. Dead cells were excluded using SYTOX Blue/Dead Cell Stain (ThermoFisher). Flow cytometric analysis was performed using an LSRFortessa X-20 Analyzer and data were analyzed with FlowJo 10.4.2 (FlowJo, LLC).

## Western blot

Testes were collected from adult male *Rad51c* mutant and wild-type mice, and protein lysates were used for immunoblotting using standard techniques. MAFs were harvested by adding hot 1x SDS running buffer onto the cells, collected and further processed for immunoblotting using standard techniques. The following antibodies were used: anti-Rad51c (clone 2H11/6, Novus Biologicals), anti-a-Tubulin (SantaCruz).

## Peripheral blood analysis

Whole blood was collected from 8–12-weeks-old mice in EDTA microcuvette tubes (BD Biosciences) and complete blood counts were analyzed using ADVIA 2120i Hematology systems (Siemens Healthineers) according to the manufacturer's instruction. Whole blood slides were stained with Diff-Quick stain.

## Colony formation unit assay

Hematopoietic colony formation unit assays were performed using bone marrow cells harvested as described above. The number of bone marrow cells was enumerated using trypan blue staining in a Neubauer chamber (Thermo Fisher). Appropriate numbers of total bone marrow cells were then exposed to various concentrations of MMC and seeded into 6-well plates with MethoCult GF M3434 (Stem Cell Technologies) media following the manufacturer's instructions.

## X-ray imaging

X-ray imaging from mouse tails was performed using an IVIS Lumina X5 imager.

## Irradiation of mice

Mice for repopulation assay received a dose of 9 Gy of total body irradiation. Additionally, mice were treated prophylactically with enrofloxacin (Baytril) in the drinking water for one week before irradiation and for 3 weeks after irradiation. $Brca2^{\Delta27/\Delta27} + Rad51c^{dah/dah}$ mice for survival analysis after irradiation received ten times 1.5 Gy of body irradiation daily.

## Competitive repopulation assay

To assess the HSC function the competitive repopulation assay was performed essentially as described previously[62]. Briefly, male bone marrow cells ($5 \times 10^4$ or $2 \times 10^5$) from $Brca2^{mut}$ and $Brca2^{\Delta27/\Delta27} + Rad51c^{dah/dah}$ mice were mixed with $5 \times 10^4$ of wild-type female bone marrow and injected into lethally irradiated (900 Gy) female wild-type recipients. Three to four recipients were used for each genotype. After 16 weeks, recipient mice were killed and genomic DNA from peripheral blood was extracted using PureLink Genomic DNA Kit (Invitrogen) according to the manufacturer's instructions. Thereafter, the relative contribution of the tested donor mutant and wild-type bone marrow to peripheral blood chimerism was assessed using quantitative PCR (qPCR) for the presence of male-specific Y-chromosome (SRY-Forward, 5′- TGTTCAGCCCTACAGCCACA-3′, and SRY-Reverse, 5′-CCTCTCACCACGG- GACCAC-3′), and detection was performed with the TaqMan probe SRY-T, and 5′-FAM– ACAATTGTCTAGAGAGCATGGAGGGCCA–BHQ1-3′. Primers for murine b-actin sequence were b- actin-Forward, 5′-ACGGCCAGGTCATC ACTATTG-3′, and b-actin-Reverse, 5′- ACTATGGCCTCAAGGAGT TTTGTCA-3′, and detected with the TaqMan probe b -actin-T, 5′-FAM–AACGAGCGGTTCCGATGCCCT–BHQ1-3′. The amplification reaction was performed using TaqMan Multiplex Mix Kit (Thermo-Fisher) according to the manufacturer's instructions.

## Whole-exome sequencing of FA patient DNAs

Library preparation and target enrichment were achieved using the SureSelect Human All Exon system of different versions (Agilent) or TruSeq DNA Exome technology (Illumina), and was followed by next-generation sequencing on a HiSeq2500 instrument (Illumina). The average exome coverage was determined using a complete list of human exons generated by the *UCSC Table Browser*. The same procedure was performed for FA gene coverage. Generally ≥80 of the reads were on target with >85% of bases covered at 10× depth. Data were analyzed using Next*GENe* Sequence Analysis software *(Softgenetics)*. Finally, a manual filtering step based on data mining was carried out to prioritize relevant mutations. A minimum coverage of 10 reads was set as the threshold for any variant to be considered faithful. The variant detection frequency was limited to a minimum of 20% of the reads covering any aberration. Potentially pathogenic variants were verified by Sanger sequencing generally using an *Applied Biosystems 3130xl* instrument. dbGAP analysis of "Etiological Investigation of Cancer Susceptibility in Inherited Bone Marrow Failure Syndromes: A Natural History Study": The data/analyses presented in the current publication are based on the use of study data downloaded from the dbGaP website, under phs001481.v1.p1, https://www.ncbi.nlm.nih.gov/projects/gap/cgi-bin/study.cgi?study_id=phs001481.v1.p1. The raw data of Supplementary Table 2 are protected and are not available due to data privacy laws.

## Whole-exome sequencing data analysis

Paired-end raw sequence reads ($2 \times 100$ bp) in fastq format were aligned to the reference genome (Mus Musculus - house mouse, mm10) using BWA mem with 31 bp seed length. The aligned BAM files are subjected to mark duplication, re-alignment, and re-calibration using Picard and GATK before any downstream analyses. In addition, the paired-end raw sequence reads ($2 \times 100$ bp) in fastq format were also aligned to the reference (Mus Musculus, mmu) V, D, J and C genes of T- or B- cell receptors directly using MiXCR to get T-cell receptor clonality. Based on the alignment results (BAM files) processed above, somatic mutations, including single-nucleotide variants (SNVs), small insertions and deletions (INDELs), and structural variants (SVs) were obtained through merging variants from multiple somatic variant

callers – MuTect, Pindel, and Manta/Strelka2. Mutations previously reported in a public database (dbSNP v137) with >1% allele frequency were removed. Next, we applied the following mutation-filtering criteria: (i) sequencing depth ≥ 20 for tumor and ≥10 for normal, (ii) tumor variant allele frequency (VAF) ≥ 5%, and normal VAF < 2%. Associations between somatic mutations and disease progression/treatment response were analyzed and visualized using Maftools. Somatic copy-number aberration (SCNA) analysis was conducted using the in-house application ExomeLyzer (a modified version of HMMcopy). The resulting normalized log2ratios were segmented using circular binary segmentation (CBS) algorithm implemented in the DNAcopy package of Bioconductor. The copy ratios of segments were then assigned to the over-lapping genes by CNTools with thresholds ($-0.3 <$ seg.mean $< 0.3$) and visualized by GenVisR. The obtained SCNAs were furthermore verified using CNVkit, and also visualized by the Python package - matplotlib.

### cBioportal analysis of breast cancer patient data

Breast cancer data sets deposited on cBioportal.org[39,40] that had information on both, *FANC* gene mutations (mutations and copy number alterations as defined by cbioportal.org, Metabric and TCGA PanCancer Atlas datasets, 3593 samples total) and overall survival data was used to create the Kaplan–Meier curves as indicated in the Figures. Patient data was grouped in patients with tumors containing no *FANC* mutation, polygenic FANC mutations (combinations of two *FANC* mutations) and monogenic *FANC* mutations using the FANC combinations identified in FA patients (Supplementary Tables 1 and S2) or any of the 253 possible *FANC* combinations when combining any two of the 23 FANC genes. cBioportal defines mutations as reported by the original study and includes non-synonymous mutations, amplifications and deletions. LOH and zygosity status was inferred from the reported allele frequency and CNA.

### DNA Fiber analysis

DNA Fiber experiments were performed as previously described[10]. Briefly, replication tracts of log-phase cells were pulse-labeled with 50 μM IdU and CldU before or after exposure to hydroxyurea, respectively, as indicated in the sketches. Cells were harvested, lysed and spread to obtain single DNA molecules on microscope slides before standard immunofluorescence with antibodies against IdU and CldU (Novus Biologicals, BD Biosciences).

### Statistical analysis

Statistical data analysis was performed using GraphPad software. Significant differences between sample groups for mouse experiments were determined using ANOVA. Statistical analysis for DNA fiber assays was determined using the Mann-Whitney test. Statistical analysis for cancer genome data uses the log-rand test and was calculated on cBioportal.org[39,40]. Statistical significance is indicated with asterisks as follows (*$p < 0.05$, **$p < 0.01$, ***$p < 0.001$, ****$p < 0.0001$, n.s. $p > 0.05$).

### Reporting summary

Further information on research design is available in the Nature Portfolio Reporting Summary linked to this article.

## Data availability

All data pertaining to the results in the manuscript are available in the main text, the Supplementary materials, and Source Data file. The raw data in Supplementary Table 2 are protected and are not available due to data privacy laws. Any additional data and material requests are required to comply with institutional policies and can be requested by contacting the corresponding authors. Source data are provided with this paper.

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

## Acknowledgements

We thank Drs. John A. Tainer, Davide Moiani, Michael Longo and Gareth Williams for sharing information and discussions on the RAD51C crystal structure, Dr. Tamara M. Haygood for advice on tail pathologies seen with X-ray, as well as the MDACC Genetically Engineered Mouse Facility (GEMF), the Small Animal Imaging Facility, the Research Histology Core Laboratory core facilities, the Advanced Cytometry & Sorting Facility at South Campus (ACSF), and the Veterinary and Comparative Pathology facility at MD Anderson Cancer Center for critical support. The work was supported by the NIEHS under award 1R01ES029680, and by CPRIT RP180463, R1312 and RP180813 (K.S.), and the FA research group at the University of Wuerzburg was supported by grants from the Schroeder Kurth Fund (D.S.). Carolina Guerrero and Poojan Shukla were supported by the CPRIT Research Training Award CPRIT Training Program (RP210028). K.S. is a Rita Allen Foundation Fellow and a CPRIT scholar in Cancer Biology (previous award R1312).

## Author contributions

K.H.T. designed and performed the experiments. D.S. performed passenger mutation analysis of FA patients and a literature search. C.K., S.R.,

and Y.C. performed the DNA fiber analysis, C.G. and P.S. together with K.S. performed cBioportal analysis, X.W and J.Z performed the bioinformatics analysis, M.O. experimentally contributed to the understanding during the development of the work. C.D. contributed to the clinical understanding during the development of the work. K.S. conceived the project, designed experiments, contributed to the literature search and the DNA fiber and bioinformatics analysis, and wrote the manuscript with input from all authors.

## Competing interests

The authors declare no competing interests.
