## [Peer Review File · Nature Communications]

Hypomorphic *Brca2* and *Rad51c* double mutant mice display Fanconi anemia, cancer and polygenic replication stressREVIEWER COMMENTS

Reviewer #1 (Remarks to the Author):

The manuscript from Tomaszowski et al. describes the contribution of polygenic stress synergy in driving the disease phenotypes and chemotherapeutic sensitivity. The authors first show that combined mutation of FANC genes is a frequent occurrence in human patients with Fanconi anaemia. This study utilizes a mouse model which combines hypomorphic alleles of Rad51c (Fanco) and Brca2 (Fancd1). The authors find that combining these two alleles leads to potentiation of existing phenotypes or phenotypes that were not observed in the single allele controls. Rad51cmutBrca2mut mice have reduced mass, skeletal abnormalities, reproductive defects, hematological abnormalities, and increased cancer pre-disposition. At the cellular level, the double mutant Rad51cmutBrca2mut (fibroblasts or bone marrow) cells exhibited DNA interstrand crosslinker hypersensitivity and chromosomal instability that is greater than in allelic controls. Finally, the authors find that polygenic FANC mutations are associated with decreased survival in breast cancer patients.

This data presented in this manuscript are of high quality convince the reader of a genetic interaction between Rad51cmut and Brca2mut alleles. The authors attribute this interaction to the role of Rad51c and Brca2 in the FA repair pathway as these proteins have other functions (not least in HR). This careful consideration is especially important given that the alleles of Rad51c and Brca2 are hypomorphic and may represent separation of function mutations. The authors should also consider existing double mutant mouse models in which potentiation of phenotype was not observed (I.e. Fanca/Fancc in Noll et al 2002) or when potentiation was observed (Fancd/Fancg in Pulliam-Leath 2010). This manuscript has robust data and makes an important contribution to the field the authors should make the connection to FA less emphatic and consider other plausible explanations for the observed interaction.

Major concern

Polygenic stress synergy. The authors assert that the simultaneous mutation of Rad51c and Brca2 reveals the new concept of polygenic stress. Firstly, we agree with the authors that they find a non-epistatic relationship between Rad51cmut and Brca2mut alleles – both in cells and at the level of the organism. However, the interpretation of this finding requires incredibly careful consideration both in interpreting the mechanism of this non-epistatic relationship but also how that relates to Fanconi anemia.

Rad51c (Fanco) and Brca2 (Fancd1) both have distinct roles in DNA repair outside of the Fanconi pathway to repair DNA interstrand crosslinks. Indeed, Rad51c the Rad51 paralog has a well established role in recombination likely through the modelling of Rad51 filaments. Brca2 also plays a significant role in HDR. These roles in HDR are not features of all components of the FA DNA repair pathway and may explain why these factors have hypersensitivity to DNA damage agents other than crosslinking agents (e.g. ionising radiation). Perhaps the clearest demonstration that these factors play important roles outside of FA mediated repair comes when one considers the phenotypes of mice that are deficient in these factors – loss of either Brca2 or Rad51c results in embryonic lethality (in the case of Rad51c it is essential for cells to survive). As the authors note this is not the case for classical Fanconi genes in which mouse models have only subtle phenotypes. One therefore, has to question if the findings relate to the role of these factors in FA or their non-FA functions in more generalised HDR.

Rad51c and Brca2 play distinct roles in HDR. Previous reports, (Jensen et al 2013) show that at the cellular level Brca2 and Rad51c are epistatic with respect to sensitivity – this is not the observation in the current manuscript. Is the non-epistasis more easily explained as a genetic interaction between Rad51c and Brca2?

As null alleles of Rad51c and Brca2 are lethal the authors have used hypomorphic alleles throughout this study. This approach is the only available one however, it does lead to some limitations and complexity in interpretation. Firstly, these hypomorphic alleles are likely to be more than just reductions in protein level - indeed Rad51c protein seems as abundant in the mutant allele as the WT (Fig1d). Whilst the chosen mutation in Rad51c is in the region of the human pathogenic R258H variant it remains unclear if a deletion in this region would have the same consequence as the R->H mutation. The c-terminal deletion of Brca2 is viable in contrast to the embryonic lethality of a Brca2 null. The c-terminal deletion is proposed to disrupt the interaction between Rad51 and Brca2. Whilst these mutations are viable, we do not understand how these mutations subvert lethality and exact consequence of the mutation on the mechanics of DNA repair. The worst case scenario here is that each allele causes some perturbation of HDR and the combination is worse than either single mutant. We simply do not understand enough about these alleles to understand why they have a non-epistatic relationship.

Do the human FA patients with Rad51c mutations have mutations of other FA genes? If not, then why do the Rad51c single mutant mice not recapitulate the FA phenotype

This major concern, critical to the central premise of the manuscript, can be dealt with through changes to the narrative of the manuscript. The authors should adopt a less Fanconi centric narrative and more fully consider the genes which are being studied.

If the authors want to address the issue of compound mutations in FA genes potentiating the phenotype it would really be necessary to focus on more "canonical" factors. I would suggest that the authors generate a double mutant Fanconi mouse between Fanca (the most commonly mutated gene in Fanconi anemia) and another canonical Fanconi anemia factor (they find interaction between Fanca and Fance, FancI and Fancg).

It should be noted that some work on this area has already been published: Firstly, Fanca/Fancc seems to cause no exacerbation of phenotype. Secondly, Fancg makes the phenotype of Fancc worse - there may be multiple explanations of this including the role of Fancg in recombination as it was identified as Xrcc9. Thirdly, the authors find mutation between Fanca and Fancm (Supplementary Table 2 and cite previous reports S Table 1). The genetic interaction between this helicase and FA core complex components at the cellular level has previously been described and it was found that loss of Fancm rescues the cellular hypersensitivity of core complex components to crosslinking agents (Mosedale et al 2005).

Minor Concerns

1. Do the authors predict loss of one Fanc factor and heterozygous mutation of another to cause a phenotype? Do they think this would be due to a haploinsufficiency? (Supplementary Table 2)

2. Is there a difference between Brca2mut and double knockout

Fig 2a

Fig 2d

Fig 2e

Fig 2i

Jensen et al BRCA2 is epistatic to the RAD51 paralogs in response to DNA damage doi: 10.1016/j.dnarep.2012.12.007

Noll et al Fanconi anemia group A and C double-mutant mice: Functional evidence for a multi-protein Fanconi anemia complex [https://doi.org/10.1016/S0301-472X\(02\)00838-X](https://doi.org/10.1016/S0301-472X(02)00838-X)

Pulliam-Leath 2010 Genetic disruption of both Fancc and Fancg in mice recapitulates the hematopoietic manifestations of Fanconi anemia doi: 10.1182/blood-2009-08-240747

Mosedale et al The vertebrate Hef ortholog is a component of the Fanconi anemia tumor-suppressor pathway DOI: 10.1038/nsmb981

Reviewer #2 (Remarks to the Author):

This is an interesting paper describing existence of co-mutations in FANC genes in FA patients, phenotype of mice with hypomorphic mutations in both Brca2 and Rad51c. The authors also described that polygenic FANC mutations (combination found in FA patients) in breast cancer are associated with lower survival in breast cancer patients.

Although the phenotype of the mice is interesting, it is not a good model to conclude "Fanconi Anemia cancer predisposition and drug sensitivity is driven by synergizing FANC mutations."

Major points:

- 1) I do not think the Brca2 and Rad51c hypomorphic mutant mouse model is a good model for testing the functional relevance of co-mutations in FANC genes in FA patients. First, the authors have not found any co-mutations in Brca2 and Rad51c in FA patients. Second, the FA patients with co-mutations seem to have biallelic mutations in one of the FA genes and a monoallelic mutation in another FA gene. In contrast, the Brca2 and Rad51c hypomorphic mutant mouse model has biallelic mutations in both genes. These situations are very different.
- 2) It is not clear whether the mutations found in breast cancer samples are biallelic or monoallelic, somatic or germline. The authors should explain this point. Otherwise it is difficult to judge whether the mice model is relevant to polygenic FANC mutations in breast cancer.

Minor points:

- 1) Extended Data Fig 1. B and C : Mislabeled.
- 2) Figure 1a and Supplemental table 2. No control (normal individual data) is shown. Supplemental table 2. Ser. # 36 If this patient is an FA-A patient, the FANCA mutations (FANCA c.702G>A, p.Met234Ile; FANCA c.2574C>G, p.(Ser828Arg)) should not be listed as "FANC passenger Mutation."
- 3) P.4 L.32 "In contrast to Rad51c KO mice, the mutant mice express the truncated protein (Fig. 1d),"  "In contrast to Rad51c KO mice, the mutant mice express the internally-deleted protein (Fig. 1d),"
- 4) Supplemental Table 3. The table should contain all the genotypes.
- 5) P.6 L.34 "mouse embryonic adult fibroblasts (MAF)"  "mouse adult fibroblasts (MAF)"
- 6) P.7 L.23 "at a lower concentration of MMC (2 mg/ml)," "at a lower concentration of MMC (2 ng/ml),"
- 7) P.7 L.38 "(Fig. 4b,c, and Fig. 7a),"  "(Fig. 4b,c, and Extended Data Fig. 7a),"
- 8) How did the author prove the clonality of the expanded T cells in the mice?

Reviewer #3 (Remarks to the Author):

NC-07990

To the Authors:

In the current manuscript, Tomaszowski et al have generated a mouse model which is

homozygous for a hypomorphic allele of Rad51C, and the mouse survives. They have also developed a mouse model which is homozygous for a hypomorphic allele of Brca2, and the mouse survives. And the resulting double homozygote (Brca2 mt/mt, Rad51c mt/mt) is viable but has a severe phenotype which is more severe than the single homozygotes. Overall, these are important results. They demonstrate that Brca2 and Rad51C, although they are both FA genes, have some non-epistatic functions. In other words, these two genes do not simply work in a linear pathway and each gene has functions outside of the canonical pathway.

If the authors had simply described this double homozygote mouse, both at an organismal level (ie, the physical and physiologic features) and the cellular level, the paper would have been far stronger. Unfortunately, the manuscript, as written, has many problems in its terminology, its rationale, and its overinterpretation of the data.

Terminology: The paper unfortunately uses several terms which are not standard terms in human or mouse genetics and is therefore very misleading to the reader. I read the abstract several times, and was confused by the terminology. You use the term "functional synergy" I believe you are referring to the non-epistatic relationship of the two genes. Also, what is polygenic stress synergy? This is not a standard term. Also, most importantly, it is not clear from the abstract that you have generated and analyzed double homozygote mutant mice (ie, Brca2 mt/mt, Rad51c mt/mt mice).

The abstract, as written, says that you have combined "hypomorphic mutations in two Fanc genes". But is not clear here that you have detected the broad spectrum of the human disease in the double heterozygote mice or the double homozygotes.

Also, throughout the paper, you refer to the Brca2 mt + Rad51C mt mice. Again, this is misleading. It would help if you refer to specific genotypes in the paper. For instance, have you analyzed the phenotypes of the Brca2 mt/mt, Rad51c mt/mt mice, the Brca2 +/+, Rad51c mt/mt mice, the Brca2 mt/mt, Rad51c +/+ mice, the Brca2 +/-, Rad51c mt/mt mice, etc etc. There are 16 genotypes to consider here from the cross.

Rationale: The paper provides two tables in the supplement showing FA cell lines, and FA patients, with a specific FA gene homozygote mutation (ie, as specific FA subtype, like FA-A) but which also carry a non-synonymous mutation in another FANC gene. While these non-synonymous mutations may encode a FANC protein with an amino acid change, there is no evidence that these proteins are dysfunctional. The proteins may simply be benign variants. So, these results do not provide a good rationale for generating the Brca2 mt/mt, Rad51c mt/mt mice. In fact, simply performing a double k/o of any two FA genes is a good idea, especially if the cross generates an interesting phenotype, as you have shown here.

Overinterpretation: Again, the term polygenic stress synergy is very misleading. What your data show, more simply stated, is that Brca2 and Rad51c cooperate in a common pathway but also have distinct non-epistatic functions outside this pathway. Therefore the double homozygote mutant mouse (Brca2 mt/mt, Rad51c mt/mt mouse) has a more severe phenotype, including some of the severe phenotypes of humans with FA. This severe phenotype appears to result from a cellular defect in replication restart (from the homozygote loss of Rad51C) and a cellular defect in replication fork protection (from the homozygote loss of Brca2).

Further analysis required:

The analysis of the MAFs in figure 3 is the most interesting section of the paper, but it requires more analysis. The MAFs with the double homozygote genotype (Brca2 mt/mt, Rad51c mt/mt) appear to have a more severe fork protection defect than the MAFs with either the Brca2 mt/mt genotype or the Rad51c mt/mt genotype. Can this cellular phenotype of the Brca2 mt/mt, Rad51c mt/mt MAFs be complemented by transfection with the cDNA encoding wt Brca2 cDNA or the wt Rad51c cDNA?

The development of spontaneous T-cell leukemias in the Brca2 mt/mt, Rad51c mt/mt model, but not in the single gene homozygous mutant mouse model, is very interesting. More analysis of these leukemias is warranted. Do the cells have a stronger FA-like phenotype, with increased chromosome breaks and translocations?

The human breast cancer analysis in Figure 5 is not well supported by the data shown. Again, even though the results from the Brca2 mt/mt, Rad51c mt/mt mouse model are very interesting on their own, I am not convinced by the connection to human breast cancer patient survival in Figure 5. This connection seems speculative. I am confused by the data in Figure 5a, and more clarification is required. The authors claim to have identified breast tumors, via cBioPortal, that have mutations in BRCA2, RAD51C, and in both genes. More characterization of the mutations in these tumors is warranted. For instance, have you identified a tumor with a pathogenic mutation in one allele of BRCA2, a pathogenic or loss (LOH) of the second BRCA2 allele, and a pathogenic or nonsynonymous mutation in one of the RAD51C alleles? If so, I would have predicted that this tumor would have a significant cellular defect and that the cancer patient might have an improved survival, not decreased survival.

Other points:

Introduction, page 3: In fact, there are explanations for the variable phenotypes observed for FA patients from the same kindred, despite their common genotypes. These explanations include modifier genes, variable teratogenic exposures during each pregnancy, and mosaicism.

RESPONSES TO REVIEWER COMMENTS

We would like to thank all the reviewers for their thorough and thoughtful review, which without a doubt greatly improved our manuscript. We have substantially revised the manuscript and added requested data additions in Fig. 3g, Fig. 4e,f,g,h,i, Supplementary Fig 1b, Supplementary Fig 5b,c, Supplementary Fig 7d, e, Supplementary Fig 9b, Supplementary Table 3, Supplementary Table 4, and Supplementary Table 5.

Please see specific points and answers below in blue.

Reviewer #1 (Remarks to the Author):

The manuscript from Tomaszowski et al. describes the contribution of polygenic stress synergy in driving the disease phenotypes and chemotherapeutic sensitivity. The authors first show that combined mutation of FANC genes is a frequent occurrence in human patients with Fanconi anaemia. This study utilizes a mouse model which combines hypomorphic alleles of Rad51c (Fanco) and Brca2 (Fancd1). The authors find that combining these two alleles leads to potentiation of existing phenotypes or phenotypes that were not observed in the single allele controls. Rad51cmutBrca2mut mice have reduced mass, skeletal abnormalities, reproductive defects, hematological abnormalities, and increased cancer pre-disposition. At the cellular level, the double mutant Rad51cmutBrca2mut (fibroblasts or bone marrow) cells exhibited DNA interstrand crosslinker hypersensitivity and chromosomal instability that is greater than in allelic controls. Finally, the authors find that polygenic FANC mutations are associated with decreased survival in breast cancer patients.

This data presented in this manuscript are of high quality convince the reader of a genetic interaction between Rad51cmut and Brca2mut alleles. The authors contribute this interaction to the role of Rad51c and Brca2 in the FA repair pathway as these proteins have other functions (not least in HR). This careful consideration is especially important given that the alleles of Rad51c and Brca2 are hypomorphic and may represent separation of function mutations. The authors should also consider existing double mutant mouse models in which potentiation of phenotype was not observed (I.e. Fanca/Fancc in Noll et al 2002) or when potentiation was observed (Fanc/Fancg in Pulliam-Leath 2010). This manuscript has robust data and makes an important contribution to the field the authors should make the connection to FA less emphatic and consider other plausible explanations for the observed interaction.

Major concern

1. Polygenic stress synergy. The authors assert that the simultaneous mutation of Rad51c and Brca2 reveals the new concept of polygenic stress. Firstly, we agree with the authors that they find a non-epistatic relationship between Rad51cmut and Brca2mut alleles – both in cells and at the level of the organism. However, the interpretation of this finding requires incredibly careful consideration both in interpreting the mechanism of this non-epistatic relationship but also how that relates to Fanconi anemia.

We apologize for not having explained these points better in the original manuscript. The term polygenic stress synergy refers to our results that show synergistic replication stress, that is more than the sum of the single gene mutation (Fig. 3c). We have now redefined the term as "polygenic replication stress" to better reflect this. We also have made changes throughout the text to better discuss this as well as its limitations, e.g. that there are accumulating non-canonical FANC functions and it will be of great interest to comprehensively unravel the specific causes and functional defects of unexpected FANC interactions in future studies"

2. Rad51c (Fanco) and Brca2 (Fancd1) both have distinct roles in DNA repair outside of the Fanconi pathway to repair DNA interstrand crosslinks. Indeed, Rad51c the Rad51 paralog has a well established role in recombination likely through the modelling of Rad51 filaments. Brca2 also plays a significant role in HDR. These roles in HDR are not features of all components of the FA DNA repair pathway and may explain why these factors have hypersensitivity to DNA damage agents other than crosslinking agents (e.g. ionising radiation). Perhaps the clearest demonstration that these factors play important roles outside of FA mediated

repair comes when one considers the phenotypes of mice that are deficient in these factors – loss of either *Brca2* or *Rad51c* results in embryonic lethality (in the case of *Rad51c* it is essential for cells to survive). As the authors note this is not the case for classical Fanconi genes in which mouse models have only subtle phenotypes. One therefore, has to question if the findings relate to the role of these factors in FA or their non-FA functions in more generalised HDR.

We have now better discussed these points in particular in the first paragraph of the introduction, and throughout the manuscript. While both *BRCA2* and *RAD51C* are proteins contributing to HDR, they are FA susceptibility genes nevertheless. As the reviewer pointed out, the contribution of HDR defects to the development of FA is elusive. *RAD51* has been shown to be required for crosslink repair in a HDR independent manner by stabilizing stalled forks (Walter Science 2011), *FANCR/RAD51* patient mutations cause fork protection defects but no HDR defects (Smogorzewska Mol Cell 2015), and restoration of FP but not ICL repair rescues HSC functions in vivo (Tong Nature comm 2018). As also pointed out by the reviewer HDR defects typically cause embryonic lethality. The hypomorphic *Brca2* and *Rad51c* alleles do not, in consistent with an HDR defect. The polygenic mutant mouse does not result in embryonic lethality. Most importantly, the mouse develops all phenotypes consistent with classical FA. Together, we think it is unlikely that the two hypomorphic mutations simply cause a synergistic HDR defect. We therefore focused on investigating replication reactions in this initial report.

However we wholeheartedly agree with the reviewer that these as well as other canonical FANC genes have well established roles outside of canonical DNA repair (*FANCG* and *FANCC* have been shown to be important during single strand annealing, *FANCD2* is important in metabolism, *FANCG* and others control mitophagy and immune signaling via mitochondrial functions to name a few) and that it is of great importance to understand the specific reason for the unexpected interaction in future studies, as we now also point out in the discussion.

3. *Rad51c* and *Brca2* play distinct roles in HDR. Previous reports, (Jensen et al 2013) show that at the cellular level *Brca2* and *Rad51c* are epistatic with respect to sensitivity – this is not the observation in the current manuscript. Is the nonepistasis more easily explained as a genetic interaction between *Rad51c* and *Brca2*? As null alleles of *Rad51c* and *Brca2* are lethal the authors have used hypomorphic alleles throughout this study. This approach is the only available one however, it does lead to some limitations and complexity in interpretation. Firstly, these hypomorphic alleles are likely to be more than just reductions in protein level - indeed *Rad51c* protein seems as abundant in the mutant allele as the WT (Fig1d). Whilst the chosen mutation in *Rad51c* is in the region of the human pathogenic R258H variant it remains unclear if a deletion in this region would have the same consequence as the R->H mutation. The c-terminal deletion of *Brca2* is viable in contrast to the embryonic lethality of a *Brca2* null. The cterminal deletion is proposed to disrupt the interaction between *Rad51* and *Brca2*. Whilst these mutations are viable, we do not understand how these mutations subvert lethality and exact consequence of the mutation on the mechanics of DNA repair. The worst case scenario here is that each allele causes some perturbation of HDR and the combination is worse than either single mutant. We simply do not understand enough about these alleles to understand why they have a non-epistatic relationship.

Fanconi Anemia patients do not contain mutations that renders protein knock-outs, since these are incompatible with life as the reviewer has noted. Functions of the C-terminal end of *BRCA2* has been quite exhaustively studied. Amongst others, it is known that it is greatly dispensable for HDR. It harbors a *RAD51* interaction site distinct from its *brc*-repeats in more N-terminal region of the gene. Specifically, this C-terminal interaction stabilizes *RAD51* filaments, which is essential for fork protection and dispensable for HDR. The latter requires the *RAD51* loading capacity of the *brc* repeats. Fanconi *FANCD1/BRCA2* patients harbor truncation mutations of the C-terminal end as present in the *Brca2* mouse model used.

The deletion mutation in the *Rad51c* mouse mutant protein is not simply just in close proximity of R258, which was found mutated in the first reported FA-O patient. Rather it is located within the same α -helix, which is an integrated structure: There are 3.6 residues per turn. An arginine is a charged amino acid, which forms salt and hydrogen bridges integral to structural conformations of proteins. A two-amino acid deletion

within an α -helix positions the charged Arginine on the opposite side of the helix, making it impossible to maintain the interactions it had in the wild-type protein conformation. So irrespective of other defects, it will cause a functional defect in the R258, because an α -helix is an integrated structural unit of a protein. To illustrate this better, we have included a helical wheel for protein projection in the Supplementary Fig 1b, illustrating the perturbed arginine position in the mutant protein.

Jensen et al solely tested cellular survival to mitomycin C by knock down of RAD51C and BRCA2 without any molecular or cellular test, presumably measuring HDR and reported epistasis with MMC. Yet, the plating efficiency is clearly reduced with the double knockdowns compared to single knockdowns, suggesting non-epistasis without exogenous stress. Lopes (Nature comm 2020) showed that in the context of replication, double knock down of BRCA2+RAD51C is not epistatic. Taken together there clearly is more to learn about these proteins and their phenotypes than epistasis during HDR using full protein knock downs. Our data suggests that hypomorphic mutations are synergistic and cause strongly augmented replication fork instability.

4. Do the human FA patients with Rad51c mutations have mutations of other FA genes? If not, then why do the Rad51c single mutant mice not recapitulate the FA phenotype

We are pleased to report that we have identified an FA patient with polygenic mutations in BRCA2 and RAD51C in the NIH dbGAP database (Supplementary Table 3). Specifically, the patient has homozygous mutations in BRCA2 V2466A, which lowers BRCA2 protein concentration, an additional heterozygous mutation in BRCA2 N372H, which is a low-penetrance variant significantly associated with an increased risk of overall cancer, and a heterozygous damaging mutation in RAD51C G264S. Interestingly, the hypomorphic *Rad51c^{dah/dah}* mutations in our mouse model is sandwiched between the originally reported RAD51C R258H mutation and this RAD51C G264S mutation within the same integrated protein structure.

The *Rad51c^{dah/dah}* mutant mice similar to many other single mutant Fanc gene mice have mild to no phenotypes under otherwise *unstressed* conditions. With external damaging agents, many Fanc mouse models show a much improved recapitulation of FA patient phenotypes. Even what is currently considered the best FA model, *Fancd2^{KO/KO} + Aldh2^{KO/KO}*, only fully expresses the phenotypes with external treatment of ethanol. While no FA patient with FANCD2 plus ALDH2 mutations was known at the time of publication of the mouse model, later FA patients in Japan were identified with biallelic FANCD2 + heterozygous *ALDH2* mutations. No patients with homozygous or biallelic mutations in *ALDH2*, as modeled and required in the mouse for phenotypic manifestation in mice, were found. This point has been explained in that the repair or stress capacity of laboratory mice is relatively larger compared to humans, requiring homozygous mutations in *ALDH2* plus the addition of ethanol. The thought is that in the human setting and outside the laboratory, heterozygous *ALDH2* mutations are sufficient to elicit- in this case- sufficient aldehyde induced stress. This point may in analogy explain why we predominantly find heterozygous co-mutations in FA patients.

Our work shows that an additional *Fanc* gene mutations in mice is sufficient to elicit a broad range of FA phenotype in mice, and that the additional *Fanc* gene mutation causes intrinsic replication stress that reveal and amplifies the replication defect of another *Fanc* mutation. Since defects in fork protection similar to defects in ICL repair drive genome instability, and since restoration of fork protection has been demonstrated to be sufficient to restore FA phenotypes even in the absence of functional ICL repair in mice (Tong Nature comm 2018), it is reasonable to suggest that the increased replication stress seen in the polygenic mouse cells could drive the FA phenotypes in these mice. Importantly, we are not excluding that there are other mechanisms that can have a similar effect, such as various endogenous or external stresses including metabolites, aldehydes, or oxidative stress. In support of this, a *Fanc^{KO/KO}+Sod1^{KO/KO}* mouse showed much improved FA modeling.

5. This major concern, critical to the central premise of the manuscript, can be dealt with through changes to the narrative of the manuscript. The authors should adopt a less Fanconi centric narrative and more fully consider the genes which are being studied. If the authors want to address the issue of compound mutations in FA genes

potentiating the phenotype it would really be necessary to focus on more “canonical” factors. I would suggest that the authors generate a double mutant Fanconi mouse between *Fanca* (the most commonly mutated gene in Fanconi anemia) and another canonical Fanconi anemia factor (they find interaction between *Fanca* and *Fance*, *FancI* and *Fancg*).

It should be noted that some work on this area has already been published: Firstly, *Fanca/Fancc* seems to cause no exacerbation of phenotype. Secondly, *Fancg* makes the phenotype of *Fancc* worse – there may be multiple explanations of this including the role of *Fancg* in recombination as it was identified as *Xrcc9*. Thirdly, the authors find mutation between *Fanca* and *Fancm* (Supplementary Table 2 and cite previous reports S Table 1). The genetic interaction between this helicase and FA core complex components at the cellular level has previously been described and it was found that loss of *Fancm* rescues the cellular hypersensitivity of core complex components to crosslinking agents (Mosedale et al 2000).

We have added an expanded discussion on existing polygenic FA mouse models to the discussion section of the paper. We agree with the reviewer that previous studies are consistent with our results and interpretation, however this previously was not fully recognized. That is, that there are non-random *Fanc* gene mutation combinations that can amplify phenotypes on the organismal level and that cannot be explained by the current view of a linear FANC pathway as imagined so far for ICL repair. Our results suggest that many of the combinations identified in FA patients are functionally non-redundant, as seen by the significantly lowered survival of cancer patients with polygenic FANC tumor mutations that were found in FA patients. Importantly, this is not observed for any or randomly picked polygenic FANC mutations (Fig. 5c). So these phenotypic effects are not obtained by combining any two *Fanc* gene KO (such as *Fanca*+ *Fancc*) but rather that there is a functional collaboration that we may not yet fully understand (such as between *Fancc*+*Fancg*). As a first attempt, we see increased intrinsic replication stress to be associated with unexpected phenotype and therefore postulate polygenic replication stress as a disease initiation model as a testable hypothesis for future studies.

Minor Concerns

1. Do the authors predict loss of one *Fanc* factor and heterozygous mutation of another to cause a phenotype? Do they think this would be due to a haploinsufficiency? (Supplementary Table 2)

In contrast to HDR and repair reactions, fork protection defects can be detected in BRCA heterozygous cells (Livingston 2011, Venkataraman Cell 2019). So heterozygous mutations clearly have the capacity to perturb replication reactions under select circumstances. Yet in mice, heterozygous mutations generally are insufficient for phenotypic expression without external damage. *Fancd2^{KO/KO} + Aldh2^{KO/KO}* requires homozygous KOs in both genes to express its FA phenotypes similar to our mouse model in the absence of other externally applied damage. Mice are held in sterile conditions, limiting environmental insults that threaten replication. In a more natural environment, exogenous stress may amplify the endogenous replication stress and overstress the system to the point where heterozygous protein gene function no longer is sufficient to protect all replication forks and phenotypic. In support of this, no homozygous FANCD2+ALDH2 patient has been identified so far, but biallelic FANCD2+ heterozygous ALDH2 mutations have, in agreement with our findings of predominantly heterozygous *FANC* co-mutations in our analysis of FA patients. Indeed, with irradiation treatment, *Brca2^{WT/Δ27} + Rad51c^{dah/dah}* mice show a decreased survival compared to *Brca2^{WT/Δ27}* or *Rad51c^{dah/dah}*, supporting the notion that increased external damage and stress too is capable of amplifying the phenotype (now added to Supplementary Fig. 9b.)

2. Is there a difference between *Brca2mut* and double knockout

Fig 2a

Fig 2d

Fig 2e

Fig 2i

Statistical significances are indicated by asterisks. There also are noticeable trends otherwise as suggested in the manuscript.

3. Jensen et al BRCA2 is epistatic to the RAD51 paralogs in response to DNA damage doi: 10.1016/j.dnarep.2012.12.007

Please see above.

4. Pulliam-Leath 2010 Genetic disruption of both *Fancc* and *Fanccg* in mice recapitulates the hematopoietic manifestations of Fanconi anemia doi: 10.1182/blood-2009-08-240747

Mosedale et al The vertebrate Hef ortholog is a component of the Fanconi anemia tumor-suppressor pathway DOI: 10.1038/nsmb981

Please see above.

Reviewer #2 (Remarks to the Author):

This is an interesting paper describing existence of co-mutations in FANC genes in FA patients, phenotype of mice with hypomorphic mutations in both *Brca2* and *Rad51c*. The authors also described that polygenic FANC mutations (combination found in FA patients) in breast cancer are associated with lower survival in breast cancer patients. Although the phenotype of the mice is interesting, it is not a good model to conclude “Fanconi Anemia cancer predisposition and drug sensitivity is driven by synergizing FANC mutations.”

Major points:

1. I do not think the *Brca2* and *Rad51c* hypomorphic mutant mouse model is a good model for testing the functional relevance of co-mutations in FANC genes in FA patients. First, the authors have not found any co-mutations in *Brca2* and *Rad51c* in FA patients.

We are pleased to report that we have identified an FA patient with polygenic mutations in *BRCA2* and *RAD51C* in the NIH dbGAP database (Supplementary Table 3). The mouse models used in our study are the best approximation to the hypomorphic mutations found in FA patients, which in humans rarely result in gene knock-outs as modeled in most FA GEMs so far. The *Brca2*^{A27/A27} mouse contains a C-terminal truncation equivalent to what is found in *FA-D1* patients. The hypomorphic *Rad51c*^{dah/dah} internal deletion is sandwiched between the originally reported *RAD51C* R258H mutation and *RAD51C* G264S (the cancer mutation identified in the FA patient in the NIH dbGAP database) within the same integrated protein structure. Thus, these hypomorphic mouse models lend themselves for testing genetic interaction between *Fancc1* and *Fanco*. We have re-written the first part of the results section to reflect these new results and clarification.

2. Second, the FA patients with co-mutations seem to have biallelic mutations in one of the FA genes and a monoallelic mutation in another FA gene. In contrast, the *Brca2* and *Rad51c* hypomorphic mutant mouse model has biallelic mutations in both genes. These situations are very different.

Currently, what is considered the best FA model is a *Fancc2*^{KO/KO} + *Aldh2*^{KO/KO} homozygous double inactivation. While no FA patient with *FANCD2*+*ALDH2* mutations was known at the time of publication of the mouse model, later FA patients in Japan were identified with biallelic *FANCD2* + heterozygous *ALDH2* mutations. No patients with homozygous or biallelic mutations in *FANCD2*+*ALDH2*, as modeled and required in the mouse for phenotypic manifestation in mice, were found. This point has been explained in that the repair or stress capacity of laboratory mice is relatively larger compared to humans, requiring homozygous mutations in *ALDH2* and for full expression of the phenotypes even additional external treatment of ethanol. The thought is that in the human setting and outside the laboratory, heterozygous *ALDH2* mutations are sufficient to elicit-

in this case- sufficient aldehyde induced stress. This point may in analogy explain why we predominantly find heterozygous co-mutations in FA patients. Indeed, when gamma irradiating the mice, the heterozygous *Brca2*^{WT/Δ27} + *Rad51c*^{dah/dah} mice show a significantly decreased survival, supporting the notion that added stress is capable of amplifying the phenotype. In this case, stress is added in the form of exogenously applied damage. We now added the data showing decreased survival with heterozygous co-mutations upon gamma irradiation as Supplementary Fig. 9b.

With external damaging agents, many *Fanc* mouse models show a much improved recapitulation of FA patient phenotypes, including *Fancd2*^{KO/KO} + *Aldh2*^{KO/KO} mice. Our work shows that an additional *Fanc* gene mutations in mice is sufficient to elicit a broad range of FA phenotype in mice, and that the additional *Fanc* gene mutation causes intrinsic replication stress that reveal and amplifies the replication defect of another *Fanc* mutation. Defects in fork protection similar to defects in ICL repair drive genome instability, and restoration of fork protection has been demonstrated to be sufficient to restore hematopoietic stem cell function even in the absence of functional ICL repair (Tong Nature comm 2018). We therefore suggest that polygenic replication stress is one possibility that drives the FA phenotypes in these mice, and should be considered also in humans. That does not exclude that there are other mechanisms that can have a similar effect, such as various endogenous or external stresses including metabolites, aldehydes, or oxidative stress. In support of this, a *Fance*^{KO/KO}+*Sod1*^{KO/KO} mouse showed much improved FA modeling.

We have added the points regarding the predominantly heterozygous co-mutations in human FA patients to the discussion.

3. It is not clear whether the mutations found in breast cancer samples are biallelic or monoallelic, somatic or germline. The authors should explain this point. Otherwise it is difficult to judge whether the mice model is relevant to polygenic FANC mutations in breast cancer.

We now include a table listing the mutations of the tumor samples as Supplemental Table 5. See above regarding heterozygous co-mutations. cBioportal defines mutations as reported by the original study and includes non-synonymous mutations, amplifications and deletions. As any of these mutations can cause functional gene defects, we remained agnostic to the type of mutation and solely assessed the genetic association of polygenic compared to monogenic *BRCA2* and *RAD51C* mutation with the overall survival of the patient as phenotypic outcome. cBioportal only in the rarest occasions has any information on germline status of a given mutation, so the assumption is that the mutations are somatic (a key difference between FA patients and cancer patients). The analysis in this figure is restricted to polygenic tumor mutations regardless of germline status.

Minor points:

1. Extended Data Fig 1. B and C : Mislabeled.

Thank you for catching this, now resolved.

2. Figure 1a and Supplemental table 2. No control (normal individual data) is shown.

Normal people would not be the proper control since we are querying co-mutations in addition to bi-allelic FANC mutations, which are absent in non-FA subjects.

3. Supplemental table 2. Ser. # 36 If this patient is an FA-A patient, the FANCA mutations (FANCA c.702G>A,

p.Met234Ile; FANCA c.2ti74C>G, p.(Ser828Arg)) should not be listed as “FANC passenger Mutation.”

Thank you for catching this, we agree and have removed it.

4. P.4 L.32 “In contrast to Rad51c KO mice, the mutant mice express the truncated protein (Fig. 1d),”  “In contrast to Rad51c KO mice, the mutant mice express the internally-deleted protein (Fig. 1d),”

Thank you for catching this, now resolved.

5. Supplemental Table 3. The table should contain all the genotypes.

Now contains all genotypes from both *Brca2*^{WT/ Δ 27} + *Rad51c*^{dah/dah} and *Brca2*^{WT/ Δ 27} + *Rad51c*^{WT/dah} intercrosses

6. P.6 L.34 “mouse embryonic adult fibroblasts (MAF)”  “mouse adult fibroblasts (MAF)”

Thank you for catching this, now resolved.

7. P.7 L.23 “at a lower concentration of MMC (2 mg/ml),”  “at a lower concentration of MMC (2 ng/ml),”

Thank you for catching this, now resolved.

8. P.7 L.38 “(Fig. 4b,c, and Fig. 7a),”  “(Fig. 4b,c, and Extended Data Fig. 7a),”

Thank you for catching this, now resolved.

9. How did the author prove the clonality of the expanded T cells in the mice?

We have now significantly expanded the analysis of the T-ALL tumors and show mono and oligoclonal expansion by T-cell receptor analysis using whole exome sequencing (Fig. 4h,i)

Reviewer #3 (Remarks to the Author):

NC-07990

To the Authors:

In the current manuscript, Tomaszowski et al have generated a mouse model which is homozygous for a hypomorphic allele of Rad51C, and the mouse survives. They have also developed a mouse model which is homozygous for a hypomorphic allele of Brca2, and the mouse survives. And the resulting double homozygote (*Brca2* mt/mt, *Rad51c* mt/mt) is viable but has a severe phenotype which is more severe than the single homozygotes. Overall, these are important results. They demonstrate that *Brca2* and *Rad51C*, although they are both FA genes, have some nonepistatic functions. In other words, these two genes do not simply work in a linear pathway and each gene has functions outside of the canonical pathway.

If the authors had simply described this double homozygote mouse, both at an organismal level (ie, the physical and physiologic features) and the cellular level, the paper would have been far stronger.

We would like to thank the reviewer for his/her thorough review and thoughtful comments. Based on this we have made substantial changes in wording, terminology, rationale throughout the manuscript (changes indicated in blue throughout the manuscript) in addition to adding substantially more data panels responding to the reviewer's requests.

1. Unfortunately, the manuscript, as written, has many problems in its terminology, its rationale, and its overinterpretation of the data. Terminology: The paper unfortunately uses several terms which are not standard terms in human or mouse genetics and is therefore very misleading to the reader. I read the abstract several times, and was confused by the terminology. You use the term “functional synergy” I believe you are referring to the non-epistatic relationship of the two genes. Also, what is polygenic stress synergy? This is not a standard term. Also, most importantly, it is not clear from the abstract that you have generated and analyzed double homozygote mutant mice (ie, *Brca2* mt/mt, *Rad51c* mt/mt mice).

Thank you for pointing out potential misconceptions and help improving the communicability and impact of the manuscript. We have revised the abstract and the terminology, and removed “functional synergy”. We further revised the term to “polygenic replication stress”, which is a term that we define in this manuscript based on the data that aside oncogene induced activation stress, also tumor suppressor mutations can augment each others replication defect and so cause replication stress. We also changed and defined the nomenclature of the mouse models to better and clearly denote the double homozygous status of the mice (*Brca2* ^{Δ 27/ Δ 27} and *Rad51c*^{dah/dah}), as described in detail Fig. 1 and the text.

2. The abstract, as written, says that you have combined “hypomorphic mutations in two Fanc genes”. But is not clear here that you have detected the broad spectrum of the human disease in the double heterozygote mice or the double homozygotes. Also, throughout the paper, you refer to the Brca2 mt + Rad51C mt mice. Again, this is misleading. It would help if you refer to specific genotypes in the paper. For instance, have you analyzed the phenotypes of the Brca2 mt/mt, Rad51c mt/mt mice, the Brca2 +/+, Rad51c mt/mt mice, the Brca2 mt/mt, Rad51c +/+ mice, the Brca2 +/-, Rad51c mt/mt mice, etc etc. There are 16 genotypes to consider here from the cross.

We have revised the mouse genes to *Brca2*^{A27/A27} and *Rad51c*^{dah/dah}, and clearly state the alleles of the mice under investigation throughout the text and figures.

3. Rationale: The paper provides two tables in the supplement showing FA cell lines, and FA patients, with a specific FA gene homozygote mutation (ie, as specific FA subtype, like FA-A) but which also carry a non-synonymous mutation in another FANC gene. While these non-synonymous mutations may encode a FANC protein with an amino acid change, there is no evidence that these proteins are dysfunctional. The proteins may simply be benign variants. So, these results do not provide a good rationale for generating the Brca2 mt/mt, Rad51c mt/mt mice. In fact, simply performing a double k/o of any two FA genes is a good idea, especially if the cross generates an interesting phenotype, as you have shown here.

We are pleased to report that we have identified an FA patient with polygenic mutations in BRCA2 and RAD51C in the NIH dbGAP database (see now Supplementary Table 3). FA patients typically do not have FANC knockouts or protein elimination as modeled in most FA GEMs so far, but contain hypomorphic mutations. The mouse models used in our study are the best approximation to the hypomorphic mutations found in FA patients at hand. The *Brca2*^{A27/A27} mouse contains a C-terminal truncation equivalent to what is found in FA-DI patients. The hypomorphic *Rad51c*^{dah/dah} internal deletion is sandwiched between the originally reported RAD51C R258H mutation and the G264S (the deleterious mutation identified in the FA patient here) within the same integrated protein structure. We have re-written the first part of the results section to reflect these new results and clarification, together now providing a strong rationale for creating the polygenic hypomorphic mouse models described here.

Fanc double knockouts have previously been performed with so far inexplicable results considering the FANC ICL pathway whereby *Fancc*^{-/-}+*Fancg*^{-/-} but not *Fancc*^{-/-}+*Fanca*^{-/-} mice showed a worsening of the otherwise mild phenotypes (Noll et al Exp Hmat. 2002 and Pulliam-Leath et al Blood, 2010). Yet these reports are consistent with our results and interpretation. That is that there are non-random *Fanc* gene mutation combinations that can amplify phenotypes on the organismal level and that cannot be explained by the current view of a linear FANC pathway as imagined so far for ICL repair. Our results further suggest that many of the combinations identified in FA patients are non-redundant, as seen by the significantly lowered survival of cancer patients with polygenic *FANC* tumor mutations that were found in FA patients. Importantly, this is not observed for any or randomly picked polygenic FANC mutations (Fig. 5c). So these phenotypic effects are not obtained by randomly combining any *Fanc* KO genes (such as *Fanca*+ *Fancc*). Rather, there are specific functional collaborations that we may not yet fully understand, and may or may not involve some or any of the increasingly diverse FANC genes functions (repair, replication, mitochondria, autophagy, lipid metabolism, inflammation, etc.). As a start in unraveling this, we see increased intrinsic replication stress to be associated with unexpected phenotype and therefore postulate polygenic replication stress as a disease initiation model as a testable hypothesis for future studies. We have added this to the discussion of the paper.

We agree that we do not know if all *FANC* mutations detected in the FA patients cause a defect or in which molecular function it causes a defect. In our opinion, one of the most impressive results of the study is the fact that neither single mutant mice even in homozygous status shows any remarkable phenotypes. As such, they would not be considered pathogenic. Rather, these are potent modifier mutations that, in the context of another mutation reveal their pathogenicity. Thus, current classification of pathogenic mutations may be insufficient for consideration of polygenic phenotypes. This principle of polygenic mutations collaborating during disease development also is being increasingly exploited in other disease backgrounds, including breast cancer susceptibility prediction. The relevance of co-mutations found in the FA patients is supported by the

non-random correlation to cancer survival data. Breast cancer patients with co-mutations in genes identified in the FA patients show a significantly lower overall survival, compared to patients with no co-mutations, or to patients with FANC co-mutations not found in FA patients.

4. Overinterpretation: Again, the term polygenic stress synergy is very misleading. What your data show, more simply stated, is that Brca2 and Rad51c cooperate in a common pathway but also have distinct non-epistatic functions outside this pathway. Therefore the double homozygote mutant mouse (Brca2 mt/mt, Rad51c mt/mt mouse) has a more severe phenotype, including some of the severe phenotypes of humans with FA. This severe phenotype appears to result from a cellular defect in replication restart (from the homozygote loss of Rad51C) and a cellular defect in replication fork protection (from the homozygote loss of Brca2).

We have defined and revised the term to “polygenic replication stress”, which is based on our results that show synergistic replication stress, which is more than the sum of the phenotypes of the single gene mutation (Fig. 3c). The proposed new concept of compounding genetic alterations is extending from the idea of oncogene induced replication stress. The data here shows that tumor suppressor mutations can cause replication stress by amplifying each other’s DNA replication instability. We defined the term in the manuscript, and also clarified in the abstract that the “data highlights a non-linear FANC tumor-suppressor pathway and the importance of added stress in addition to a biallelic FANC mutation in the development of FA.”

Further analysis required:

5. The analysis of the MAFs in figure 3 is the most interesting section of the paper, but it requires more analysis. The MAFs with the double homozygote genotype (Brca2 mt/mt, Rad51c mt/mt) appear to have a more severe fork protection defect than the MAFs with either the Brca2 mt/mt genotype or the Rad51c mt/mt genotype. Can this cellular phenotype of the Brca2 mt/mt, Rad51c mt/mt MAFs be complemented by transfection with the cDNA encoding wt Brca2 cDNA or the wt Rad51c cDNA?

We have added these data in Fig. 3i and Supplementary Fig. 5b and show that transient expression of Rad51c partially rescues fork protection defects. Due to the gene size, transient *Brca2* cDNA complementation is technically not possible.

6. The development of spontaneous T-cell leukemias in the Brca2 mt/mt, Rad51c mt/mt model, but not in the single gene homozygous mutant mouse model, is very interesting. More analysis of these leukemias is warranted. Do the cells have a stronger FA-like phenotype, with increased chromosome breaks and translocations?

As requested, we significantly expanded our analysis on the T-cell leukemia (Fig. 4e, f, g, i, Supplementary Fig 7d, e) and include further hematological characterization, T-cell clonality and genome stability analysis. Taken together, the data shows the nature of lymphoblastic T-cell leukemia consistent with FA-D1, that is leukemias with high genome instability and early clonal expansion of T-cells.

7. The human breast cancer analysis in Figure ti is not well supported by the data shown. Again, even though the results from the Brca2 mt/mt, Rad51c mt/mt mouse model are very interesting on their own, I am not convinced by the connection to human breast cancer patient survival in Figure ti. This connection seems speculative. I am confused by the data in Figure tia, and more clarification is required. The authors claim to have identified breast tumors, via cbioPortal, that have mutations in BRCA2, RAD51C, and in both genes. More characterization of the mutations in these tumors is warranted. For instance, have you identified a tumors with a pathogenic mutation in one allele of BRCA2, a pathogenic or loss (LOH) of the second BRCA2 allele, and a pathogenic or nonsynonymous mutation in one of the RAD51C alleles? If so, I would have predicted that this tumor would have a significant cellular defect and that the cancer patient might have an improved survival, not decreased survival.

We have added Supplementary Table 5, which lists the tumor mutations in detail, and added more analysis on LOH and homo-/heterozygosity status. We further clarified in the text that the mutations considered are those as defined by cBioportal, which includes non-synonymous mutations, amplifications and deletions as filtered by the original studies. As any of these mutations can cause functional protein defects, we stayed agnostic to the type of mutation during the analysis. Synonymous mutations are not considered. Just as common to any other TCGA and genome analysis, we performed an association study of gene mutations with overall survival, which are reported in the results.

One of the most powerful results of the study shows that while a mutation on its own can be benign and result in no or insignificant phenotypes over a lifespan, it turns strongly disease driving when in the context of another mutation. In this context, it is important not to dismiss so far considered benign mutations, that on their own do not cause disease, and so called “passenger mutations”. As to the overall survival, whether increased genome instability is increasing or decreasing a patient’s survival is greatly dependent on the tissue of the cancer, the standard of care treatments, and the time at which the tumors were sequenced. Specifically, increased genome instability can make standard of care treatments more effective, as seen in ovarian cancer where BRCA2 carriers initially respond well to standard of care cisplatin drugs, reflected in an increased overall survival (e.g see Pal et al, Fam. Cancer 2007). Genome instability however is a double-edged sword and while it initially sensitizes to DNA damaging agents, it increases metastatic and resistance development, which results in faster death. This becomes more apparent in slower growing cancers that are treated multiple times, such as breast cancer. Consistently, breast patient carriers with BRCA mutations show a decreased overall survival (e.g. see Budroni et al, BMC cancer 2010).

Other points:

Introduction, page 3: In fact, there are explanations for the variable phenotypes observed for FA patients from the same kindred, despite their common genotypes. These explanations include modifier genes, variable teratogenic exposures during each pregnancy, and mosaicism.

We have added these points to the text.

REVIEWER COMMENTS

Reviewer #1 (Remarks to the Author):

This revised manuscript from Tomaszowski et al. describes the genetic interaction between mutant alleles of Rad51c and Brca2. The data presented is of high quality, well controlled and very exciting. It will be of great interest to the Fanconi community to know that hypomorphic Rad51c and Brca2 alleles genetically interact and precipitate a Fanconi-like phenotype in mouse. Furthermore, such "poly" mutations are observed in human breast cancer – altering survival.

However, the narrative of this manuscript remains much too Fanconi centric. As outlined below this leads to a host of issues in establishing the concept of "polygenic replication stress concept" and linking this to the human disease Fanconi anemia. These exciting results do not need to be dressed in this way. Rad51c and Brca2 are important factors and Brca2 a critical tumour suppressor factor. Simply presenting the data about Brca2 and Rad51c is of sufficient interest without trying to develop the concept of "polygenic replication stress". A detailed discussion of all possible explanations for the non-epistatic genetic interaction should be included.

Major

1. "polygenic replication stress concept"

The authors have somewhat modified the original terminology however, it brings several new issues that need to be addressed and does not counteract these initial concerns. In the abstract this concept is defined as "the occurrence of a distinct second gene mutation amplifies and drives endogenous replication stress, genome instability and disease".

a) This description is unclear and unnecessary - the interaction that the authors are describing is simply a non-epistatic genetic interaction between the Rad51c and Brca2 alleles. There is no need for the invention of new terminology to describe a classical genetic interaction.

b) As the authors discuss non-epistatic interactions have been observed in Fanconi anemia mouse models before. In the case of double Fancc and Fancg mutant mice, the phenotype was much more severe than either single mutant, more closely resembled human patients with FA. Therefore, this is not a new concept that requires a new term. (The authors now link it to a mechanism which presents new issues see below in c)

c) The term used in the revised manuscript includes the phrase "replication stress" which suggests mechanistic insight to explain this non-epistatic genetic interaction. This is an over-interpretation of the data presented whilst other explanations exist.

The authors propose that the combined mutations (Rad51cdah/dahBrca2d27/d27) result in "strong DNA replication instability" (discussion, F3A-C, S5). However, the authors go on to draw highly speculative conclusions

i) Firstly, that this establishes the concept of polygenic replication stress, analogous to oncogene-induced replication stress

This is over-interpreted. Both single mutants have replication defects (F3A) and double mutant has a worse phenotype (F3A). This suggests that two factors required for normal replication have non-redundant functions – a classical genetic interaction. Given the identity of Brca2 and Rad51c this is very interesting but it goes much too far to conclude a mechanism from the data presented.

Both Brca2 and Rad51c have functions in multiple processes (FA repair, HDR and replication fork protection). The alleles used in this study are both hypomorphic – the phenotypes could be due to some separation of function due to mutation or could be due to a reduction in the level of functional protein. The authors present data suggesting that the Rad51cdah allele has a two amino acid loss in the same helix as for which a point mutation has been described to cause FA. Whilst this is a smoking gun this is insufficient to conclude that the phenotypes of both mutants will be the same. We do not know enough about these alleles to understand the basis of the non-epistatic interaction (i.e. FA, HDR or fork protection). Taken together, I think there is insufficient evidence to conclude

that the observed phenotypes are due to replication stress.

ii) Secondly, the final line of the abstract implies that this mechanism explains the exacerbated phenotype of Rad51cdah/dahBrca2d27/d27 compared to the single mutants.

The authors show in F3D that Rad51cdah/dahBrca2d27/d27 cells are more sensitive to the DNA damaging agent MMC than either of the single mutants. Therefore, the increased cellular hypersensitivity to crosslinking agents provides an equally plausible explanation for the more severe phenotype observed in Rad51cdah/dahBrca2d27/d27 mice. It is plausible that this could be due to non-redundant roles of Rad51c and Brca2 in maintaining cellular resistance to these damaging agents (e.g. repair of the lesion). Indeed, DNA crosslinker sensitivity is the hallmark of Fanconi anemia. Given the ambiguity of the mechanism it seems inappropriate to draw the conclusion that replication stress is the driver in this case.

d) The use of Rad51c and Brca2 are problematic models due to the lethality of the null alleles leading to the complexity of analysing hypomorphs. This is compounded as these mutants are then combined, making interpretation even more complex. The divergence in phenotype of Rad51c and Brca2 nulls from other Fanconi mouse models (Fanca, Fancc, Fancd2, Fanci, Fancm, ...) does hinder generalising the conclusions in this manuscript – none of the other null alleles are lethal. It is entirely likely that this interaction is specific to Rad51C and Brca2. In other words, this is an exception to the rule rather than the rule itself.

e) The data on several occasions does not show a significant difference or the difference has not been tested between Brca2d27/d27 and Rad51cdah/dahBrca2d27/d27. (In most assays Brca2d27/d27 shows the greatest magnitude of phenotype of the single mutants).

F2A

F2D

F2E

F2I

F3A-C

2. Compound mutations driving FA

The authors argue that given the frequency of pathogenic FA alleles, the observed incidence of Fanconi anemia is lower than expected. Whilst this could represent the need for compound mutations there are other likely contributing factors. In recent years it has become evident that there is greater heterogeneity in patient presentation than first thought. Whilst many patients present with the classical FA features, a proportion of patients present only in adulthood. Therefore estimations of frequency may be an underestimate. If this issue is to be discussed other explanations should be offered as the authors do not directly ascertain an explanation.

This study has employed a double mutant mouse model (Rad51cdah/dahBrca2d27/d27). However, the Brca2/Rad51c human patient has biallelic mutation in Brca2 but is heterozygous for Rad51C (S table 3). Indeed, for other complementation groups one FA gene is biallelically mutated and the other gene contains a monoallelic change (S Table 1 and 2). It is hard to reconcile why this would cause a phenotype in human but not in mouse.

The authors make a very fair argument that FA mouse models do not recapitulate the human disease. The issue with this argument is the difficulty to explain why one wild type allele in mouse can suppress the phenotype but is insufficient in human.

They argue that this is similar to what is seen in Aldh2/Fancd2 double deficient mice i.e. that a FA phenotype of Fancd2 deficient mice is only observed when Aldh2 is inactivated. There is a clear analogy there – that there is a genetic interaction between Aldh2 and Fancd2. However, as discussed in point 1c the mechanism proposed here is unclear. (Arguably, the mechanism for the interaction between Aldh2 and Fancd2 is equally unclear).

Reviewer #2 (Remarks to the Author):

The authors have addressed most of my concerns, but there still remain some minor issues.

1) Supplemental Table 2. the legend says "37 patients", but only 36 (not 37) patients are listed in Supplemental Table 2. Please fix this issue.

2) P.5 L.26 "In contrast to Rad51c KO mice, the mutant mice express the truncated protein (Fig. 1e),"  "In contrast to Rad51c KO mice, the mutant mice express the internally-deleted protein (Fig. 1e),"

3) In Supplementary Table 4a, "p=0.0001" is shown, and in Supplementary Table 4b, "p=0.01" is shown, but in the main text (P.5 L.29.) it is written "p = 0.02; Supplementary Table 4 ". This is confusing. Please clarify this.

4) P.5 L.15 "(Fig. 1m, lower panels, and Fig. 2b,c)"  "(Fig. 1m, lower panels, and Supplementary Fig. 2b,c)."

Reviewer #3 (Remarks to the Author):

No additional comments for the authors.

Answers to the reviewers are indicated in blue font succeeding each of the reviewer's points.

Reviewer #1 (Remarks to the Author):

This revised manuscript from Tomaszowski et al. describes the genetic interaction between mutant alleles of Rad51c and Brca2. The data presented is of high quality, well controlled and very exciting. It will be of great interest to the Fanconi community to know that hypomorphic Rad51c and Brca2 alleles genetically interact and precipitate a Fanconi-like phenotype in mouse. Furthermore, such "poly" mutations are observed in human breast cancer – altering survival.

However, the narrative of this manuscript remains much too Fanconi centric. As outlined below this leads to a host of issues in establishing the concept of "polygenic replication stress concept" and linking this to the human disease Fanconi anemia. These exciting results do not need to be dressed in this way. Rad51c and Brca2 are important factors and Brca2 a critical tumour suppressor factor. Simply presenting the data about Brca2 and Rad51c is of sufficient interest without trying to develop the concept of "polygenic replication stress". A detailed discussion of all possible explanations for the non-epistatic genetic interaction should be included.

We would like to thank the reviewer for his/her supportive words and are pleased to see he/she sees our data as high quality, well controlled and most importantly very exciting. Please see below more on added discussions to address the reviewer's remaining concerns.

Major

1. "polygenic replication stress concept"

The authors have somewhat modified the original terminology however, it brings several new issues that need to be addressed and does not counteract this initial concern. In the abstract this concept is defined as "the occurrence of a distinct second gene mutation amplifies and drives endogenous replication stress, genome instability and disease".

a) This description is unclear and unnecessary - the interaction that the authors are describing is simply a non-epistatic genetic interaction between the Rad51c

and Brca2 alleles. There is no need for the invention of new terminology to describe a classical genetic interaction.

In classic genetics, non-epistatic interactions are either additive or synergistic, which is more than the simple addition of the phenotypes' severity. The phenotypes elicited by the polygenic mutant mouse are not additive but synergistic to an extent that was entirely unexpected. All known functions of BRCA2 and RAD51C, and any FA gene for that matter, ultimately culminate in suppressing DNA replication stress. Polygenic replication stress is a specific term that is related to but goes beyond the conventional and unspecific term of the genetic interaction. While in the reviewer's opinion there may not be a need for a term in this manuscript, we believe that it will be helpful in the long-term for readers and the scientific community and therefore opt to keep it in the manuscript.

b) As the authors discuss non-epistatic interactions have been observed in Fanconi anemia mouse models before. In the case of double Fancc and Fancg mutant mice, the phenotype was much more severe than either single mutant, more closely resembled human patients with FA. Therefore, this is not a new concept that requiring a new term. (The authors now link it to a mechanism which presents new issues see below in c)

As with most new concepts, observations and single reports that at the time are inexplicable precede and reinforce a new discovery. While there was a report of a Fancg knock-out worsening the phenotypes of a Fancc mouse, it was unexplained as the two genes solely were considered as part of a linear pathway during DNA crosslink repair. Our data and new concept provide new insight. We present a mouse model that shows synergism beyond a simple additive or worsening of phenotypes seen in the single mutant gene mice, along with complementary data in FA patients. We further apply the new insight to breast cancer patients data reinforcing the new concept to go beyond an interesting but at the time inexplicable observation of simply put more severe phenotypes.

c) The term used in the revised manuscript includes the phrase "replication stress" which suggests mechanistic insight to explain this non-epistatic genetic interaction. This is an over interpretation of the data presented whilst other explanations exist.

The authors propose that the combined mutations (Rad51cdah/dahBrca2d27/d27) result in "strong DNA replication instability"

(discussion, F3A-C, S5). However, the authors go on to draw to highly speculative conclusions

i) Firstly, that this establishes the concept of polygenic replication stress, analogous to oncogene-induced replication stress

This is over-interpreted. Both single mutants have replication defects (F3A) and double mutant has a worse phenotype (F3A). This suggests that two factors required for normal replication have non-redundant functions – a classical genetic interaction. Given the identity of Brca2 and Rad51c this is very interesting but it goes much too far to conclude a mechanism from the data presented.

Both Brca2 and Rad51c have functions in multiple processes (FA repair, HDR and replication fork protection). The alleles used in this study are both hypomorphic – the phenotypes could be due to some separation of function due to mutation or could be due to a reduction in the level of functional protein. The authors present data suggesting that the Rad51cdah allele has a two amino acid loss in the same helix as for which a point mutation has described to cause FA. Whilst this is a smoking gun this is insufficient to conclude that the phenotypes of both mutants will be the same. We do not know enough about these alleles to understand the basis of the non-epistatic interaction (i.e. FA, HDR or fork protection). Taken together, I think there is insufficient evidence to conclude that the observed phenotypes are due to replication stress.

The term “polygenic replication stress” does not imply mechanistic insight on a molecular level. In comparison, after many years of studying a phenomenon termed “oncogene induced replication stress” it is still not fully understood how oncogene expression causes replication stress. However, the known molecular functions of FA proteins ultimately result in replication stress, whether it is their direct role in repair, replication or more unexplored roles in metabolism. Moreover, we provide a new concept as a “testable principle” (end of discussion) rather than providing one detailed molecular mechanism or function for a complex disease. We now also added this at the end of the abstract to ameliorate the reviewer’s concerns.

ii) Secondly, the final line of the abstract implies that this mechanism explains the exacerbated phenotype of Rad51cdah/dahBrca2d27/d27 compared to the single mutants.

The authors show in F3D that Rad51cdah/dahBrca2d27/d27 cells are more sensitive to the DNA damaging agent MMC than either of the single mutants. Therefore, the increased cellular hypersensitivity to crosslinking agents provides an equally plausible explanation for the more severe phenotype observed in Rad51cdah/dahBrca2d27/d27 mice. It is plausible that this could be due to non-redundant roles of Rad51c and Brca2 in maintaining cellular resistance to these damaging agents (e.g. repair of the lesion). Indeed, DNA crosslinker sensitivity is the hallmark of Fanconi anemia. Given the ambiguity of the mechanism it seems inappropriate to draw the conclusion that replication stress is the driver in this case.

We now include a more balanced discussion and added that “FA genes act in many biological processes, and most notably during various DNA repair processes, so the phenotypes in the mice could in principle stem from defects other than those involving their direct function in replication reactions. Nevertheless, DNA damage ultimately too results in DNA replication stress and instability”, supporting this concept. We also have discussed other examples of added stress promoting more encompassing FA phenotypes, including polygenic Sod1+Fancc mice, where presumably the increased ROS due to Sod1 deletion promotes the more severe phenotypes, and Aldh2-/Fancd2- mice, where presumably the decreased aldehyde detoxification promotes the more severe phenotypes. The aforementioned sources of stress, ROS and aldehydes, are not repaired by the same pathways, but both lead to replication stress, suggesting DNA replication stress a common denominator for FA disease severity.

d) The use of Rad51c and Brca2 are problematic models due to the lethality of the null alleles leading to the complexity of analysing hypomorphs. This is compounded as these mutants are then combined, making interpretation even more complex. The divergence in phenotype of Rad51c and Brca2 nulls from other Fanconi mouse models (Fanca, Fancc, Fancd2, Fanci, Fancm, ...) does hinder generalising the conclusions in this manuscript – none of the other null alleles are lethal. It is entirely likely that this interaction is specific to Rad51C and Brca2. In other words, this is an exception to the rule rather than the rule itself.

Both BRCA2 and RAD51C are FA genes. FANCD1 (BRCA2) and FANCO (RAD51C) mutations in FA patients are not nulls, and neither are other “canonical” FA mutations. Most FA patients contain hypomorphic mutation, whereby the Brca2 hypomorphic mutation we use in our model is virtually identical to those found in some FA patients, and the Rad51c mutation is a best approximation by affecting the same structural unit of the protein as the patient mutation does. Importantly, just as

the patient mutations, the proteins are expressed and thus support the survival of the patients, mice and cells. We have identified an FA patient with polygenic *BRCA2+RAD51C* mutations, making the combination of the two reasonable.

Attempting to model a disease using a gene knockout makes a preemptive assumption that a gene has one function. We now know that this is not the case for almost all FA genes. Consistently, canonical Fanc gene KO does not recapitulate FA in mice. On the other hand, polygenic gene mutations such as *Fancc KO+ Fancg KO*, *Sod1 KO+ Fancc KO*, *Aldh2+Fancd2 KO*, better model the disease, which also is consistent with the patient data that we present here, and supporting the insight deduced from our study of a broad importance and functional consequence of polygenic FANC mutations. We discuss this in the 5th paragraph of the discussion on p.11. Our data and concept moreover is not mutually exclusive with the current dogmas of the field which sees bi-allelic FA mutations to suffice for disease development: we suggest in our discussion that the disease requires replication stress in addition to a biallelic mutation. This can be achieved by external damage and stress, or by internal factors such as ROS or aldehyde stress. Polygenic replication is another added means of achieving the additional stress required to elicit the disease.

e) The data is several occasions does not show a significant difference or the different has not been tested between *Brca2d27/d27* and *Rad51cdah/dahBrca2d27/d27*. (In most assays *Brca2d27/d27* shows the greatest magnitude of phenotype of the single mutants).

F2A

F2D

F2E

F2I

F3A-C

In all cases and tests we see the greatest magnitude of phenotype with the polygenic *Brca2+Rad51c* mutant mice, and not with the single *Brca2* mutant mice or cells. Of note, the differences between polygenic and single mutant *Brca2* cells becomes smaller when the cells are challenged with external DNA damaging/stressing reagents, supporting the overall concept that the additional mutation (in *Rad51c*) acts as an endogenous DNA stressor. We have added this point for further clarification to the second paragraph of the discussion. In the case of the blood analysis, while the polygenic mice show the most defective phenotypes, there is not always statistical significance, which is why we go on to the much more sensitive

bone marrow transplantation assays, that clearly shows the most severe functional HSC defect in the *Brca2+Rad51c* mutant mice. This severity in blood phenotypes is likely somewhat masked in the blood analysis by the restrictions of the remarkable overall phenotype of the mice: all mice succumb ~120 days of age, forcing us to restrict the blood analysis to very young mice (when they are not leukemic). Thus, the severe HSC and consequently blood phenotype is masked by the strong leukemia phenotype, similar to what is seen in the *Aldh2+Fancd2* mice, which only late in life show bone marrow failure when select mice they do not develop malignancies. The bone marrow transplantation assay shows the severity of the blood phenotype. For clarity in the figures, we do not indicate when there is no statistically significant difference in Fig. 2. We now also have added all statistically significant differences in Fig. 3.

2. Compound mutations driving FA

The authors argue that given the frequency of pathogenic FA alleles, the observed incidence of Fanconi anemia lower than expected. Whilst this could represent the need for compound mutations there are other likely contributing factors. In recent years it has become evident that there is greater heterogeneity in patient presentation than first thought. Whilst many patients present with the classical FA features, a proportion of patients present only in adulthood. Therefore estimations of frequency may be an underestimate. If this issue is to be discussed other explanations should be offered as the authors do not directly ascertain an explanation.

We agree with the reviewer and now added that “In recent years it has become evident that there is a proportion of patients that present only in adulthood, which may have skewed these estimates to some extent. On the other hand, the requirement for stress factors in addition to a biallelic *FANC* mutations too could account in part for this discrepancy.”

This study has employed a double mutant mouse model (*Rad51cdah/dahBrca2d27/d27*). However, the *Brca2/Rad51c* human patient has biallelic mutation in *Brca2* but is heterozygous for *Rad51C* (S table 3). Indeed, for other complementation groups one FA gene is biallelic mutated and the other gene contains a monoallelic change (S Table 1 and 2). It is hard to reconcile why this would cause a phenotype in human but not in mouse.

The authors make a very fair argument that FA mouse models do not recapitulate the human disease. The issue with this argument is the difficulty to explain why one wild type allele in mouse can suppress the phenotype but is insufficient in human.

They argue that this is similar to what is seen in *Aldh2*/*Fancd2* double deficient mice i.e. that a FA phenotype of *Fancd2* deficient mice is only observed when *Aldh2* is inactivated. There is a clear analogy there – that there is a genetic interaction between *Aldh2* and *Fancd2*. However, as discussed in point 1c the mechanism proposed here is unclear. (Arguably, the mechanism for the interaction between *Aldh2* and *Fancd2* is equally unclear).

We agree with the reviewer that there is still much to be uncovered to fully understand this complex disease. The analogy of our mouse model with the *Aldh2*+*Fancd2* mice goes beyond the need for polygenic homozygous gene inactivation. While at the time of the *Nature* publication by the Patel group no FA patient with both, biallelic *FANCD2*+*ALDH2* mutations was known, later a Japanese FA patient was identified who had biallelic *FANCD2*+ heterozygous *ALDH2* mutations. This was seen to validate the genetics of the mouse Patel mouse model, despite the fact that the mouse requires both homozygous *Fancd2*+ homozygous *Aldh2* KO to show the phenotypes. As argued by the field, mice have a larger repair capacity compared to humans in addition to lower external stress due to their sterile living environment, necessitating homozygous inactivation of *Aldh2* in mice to elicit the same phenotypes as in the FA patient with heterozygous mutations. This view is experimentally supported by our data in Supplementary Fig. 9b, that shows phenotypically greater severity in the heterozygous *Brca2*+ homozygous *Rad51c* mutant mice with additional external stress such as gamma irradiation. Future studies beyond the scope of this study are expected to provide further clarification.

Reviewer #2 (Remarks to the Author):

The authors have addressed most of my concerns, but there still remain some minor issues.

1) Supplemental Table 2. the legend says “37 patients “, but only 36 (not 37) patients are listed in Supplemental Table 2. Please fix this issue.

Thank you for catching this. It is now fixed.

2) P.5 L.26 “In contrast to Rad51c KO mice, the mutant mice express the truncated protein (Fig. 1e),”  “In contrast to Rad51c KO mice, the mutant mice express the internally-deleted protein (Fig. 1e),”

Thank you, now reads “internally-deleted”.

3) In Supplementary Table 4a, “ $p=0.0001$ ” is shown, and in Supplementary Table 4b, “ $p=0.01$ ” is shown, but in the main text (P.5 L.29.) it is written “ $p = 0.02$; Supplementary Table 4 ”. This is confusing. Please clarify this.

Thank you for catching this error that we missed in updating, when we added much more of breeding data in the last revised version. It is now fixed and reads “ $p = 0.01$ and 0.001 , respectively”

4) P.5 L.15 “(Fig. 1m, lower panels, and Fig. 2b,c)”  “(Fig. 1m, lower panels, and Supplementary Fig. 2b,c).”

Thank you, now fixed.

Reviewer #3 (Remarks to the Author):

No additional comments for the authors.